# Multi-objective dung beetle optimization algorithm: A novel algorithm for solving complex multi-objective optimization problems

**Wenxing Wu** **[1], Liqin Tian[1,2]\*, Junyi Wu[1], Lianhai Lin[1]**

1 School of Computer Science, Qinghai Normal University, Xining, Qinghai, China, 2 School of Computer Science, North China Institute of Science and Technology, Langfang, Hebei, China

\* tianliqin@ncist.edu.cn

## Abstract

Many increasingly complex multi-objective optimization problems are emerging, and there is an urgent need to develop new multi-objective optimization algorithms to meet the challenges. This study introduces the Multi-Objective Dung Beetle Optimization Algorithm (MODBO), which integrates competitive and neighborhood mechanisms to tackle such problems, Thanks to the dung beetle optimization algorithm's fast convergence and robust optimization finding ability in single-objective optimization algorithms. The introduction of non-dominated sorting allows the Dung Beetle Optimization Algorithm to solve multi-objective optimization problems (MOPs). To make the Dung Beetle Optimization Algorithm maintain good search ability in searching, we introduce a Competition mechanism to guide the particles' global optimal search and a Neighborhood mechanism to guide the particles' local optimal value search. An external archive is introduced to make each generation positionally optimal. Finally, to analyze whether the MODBO algorithm's improved strategy is effective, a comparison with the nine algorithms on CEC2020 was made, and the 3D sensor deployment problem was used to demonstrate that the MODBO algorithm can solve realistic problems.

## 1. Introduction

Optimization is one of the cornerstone challenges in both theoretical and practical computer science, boasting many applications across scientific disciplines and everyday scenarios. The key to the optimization problem is to find the best variable from a set of variables that satisfies the solution of a function. Many metaheuristic algorithms are currently proposed for solving single objective (SO) optimization problems. In contrast, many real-world problems involve multiple optimization goals—often two, three, or even more. When a problem has multiple objectives that must be optimized simultaneously, it is classified as a multi-objective optimization problem. To solve these problems, we

**Data availability statement:** The data and code for this study can be downloaded at:https://github.com/wu457120325/MIDBO_PLOSONE.

**Funding:** This work was supported by the Hebei Internet of Things Monitoring Technology Innovation Center(21567693H), the Qinghai IoT Key Laboratory (No.2017-ZJ-Y21) and the Fundamental Research Funds for the Central Universities (3142021009). The funders had no role in study design, data collection and analysis, decision to publish, or preparation of the manuscript.

**Competing interests:** The authors have declared that no competing interests exist.

need to use specific tools [1]. In this context, numerous powerful algorithms have been proposed. Many algorithms have been proposed and work well in different problems. But the "No Free Lunch" theorem says no algorithm is the best for all problems.

With the increase in problem complexity and computational power, researchers have developed various multi-objective algorithms to address various optimization challenges. These algorithms can be categorized into several main classes based on their fundamentals, search mechanisms, and application strategies: classical methods, evolutionary algorithms, population-based intelligent algorithms, and hybrid and integrated methods. Each algorithm type has unique advantages and scope of application, and choosing the appropriate algorithm is crucial for effectively solving multi-objective optimization problems.

Classical methods are based on mathematical planning theory and are suitable for problems with clear mathematical expressions or constraints. Such methods deal with the relationship between multiple objectives by transforming them into a single comprehensive objective or a specific strategy. For example, the weighted summation method [2] and the ε-constraint method [3]. However, this method has some drawbacks: its application scope is limited; it is difficult to deal with nonlinear, discrete, or complex structure problems, especially those objective functions that cannot be expressed analytically; it is easy to fall into local optima; which can prevent algorithms from discovering the global Pareto frontier. This issue often arises due to a lack of effective global search capabilities. Additionally, methods like the weighted summation approach require the manual setting of weights, which can introduce subjective bias and make it difficult to find all Pareto-optimal solutions.

Evolutionary Algorithms (EAs) are inspired by natural selection and genetics. They can handle complex, nonlinear, and multimodal multi-objective problems without needing a specific mathematical structure, making them very flexible and robust for real-world challenges. Typical evolutionary algorithms include NSGA-II (Non-dominated Sorting Genetic Algorithm II) [4], PESA-II (Pareto Envelope-based Selection Algorithm II) [5], and SEPA2 (Self-Adaptive Evolutionary Pareto Archive Algorithm 2) [6], and DE (Differential Evolution, DE) [7]. This method is computationally expensive, especially for high-dimensional or multi-objective problems; it requires many iterations to converge to a satisfactory solution set.

The third type of algorithm is population-based intelligent algorithms. The behavior of groups in nature inspires these algorithms, such as bird flocks, ant colonies, and bee colonies, and are used to solve multi-objective optimization problems. Examples include particle swarm optimization [8], ant colony optimization [9], and artificial bee colony algorithms [10]. The advantage of this approach is that it exploits the intelligence of group behaviors in nature and can efficiently search for solutions in complex environments. The algorithm possesses good parallelism, and the individuals in the group can search in parallel, improving the algorithm's efficiency. It has a strong adaptability ability, which can dynamically adjust the search strategy according to environmental changes and is suitable for dynamic optimization problems. The drawback of the method is that if the population concentrates on a certain region too early, it may lead to early termination of the algorithm and miss a better solution.

The last one is based on hybrid and integrated methods, combining multiple algorithms' advantages to solve complex problems more efficiently. Enhancing global search through local search can speed up convergence and improve the quality of the solution. It can also flexibly combine different algorithms and techniques according to the characteristics of the problem and is highly adaptable. The disadvantage is that designing and realizing hybrid algorithms tends to be more complex than single algorithms, increasing the difficulty of development and debugging.

Compared with the other three algorithms, the hybrid and integrated approach is more practical in practice, and most of the current heuristic algorithm improvements are based on this approach. To design such algorithms, we need to find a suitable heuristic algorithm and improve it with multiple objectives first, then evaluate its performance after improvement to improve it further.

Typically, the performance of a multi-objective algorithm depends on the effectiveness of the original single-objective algorithm. The better the original algorithm performs, the better the potential for the multi-objective version. There are several ways to improve single-objective algorithms for multi-objective optimization. NSGA-II is a genetic algorithm that generates a Pareto frontier by introducing a nondominated sort. This means it finds a set of solutions where no single solution is better than the others in all objectives. MOEA/D decomposes the multi-objective problem into several single-objective subproblems. MOEA/D uses an external archive to store non-dominated solutions. These solutions guide the search direction, helping to generate a diverse set of Pareto-optimal solutions. In recent years, most algorithms have been extensions of the particle swarm algorithm, so most algorithm improvement strategies follow those of MOPSO (Multi-Objective Particle Swarm Optimization). Yannibelli [11] combines multi-objective annealing algorithms with a multi-objective technique and demonstrates the algorithm's performance on a multi-objective scheduling problem, In the early stage of the algorithm, the simulated annealing algorithm is used to fine-tune the solution. When the solution starts to converge, it further explores potential solutions. Tawhid [12] proposed a multi-objective sine-cosine algorithm(MOSCA), introduced non-dominated ordering and congestion distance in the sine-cosine algorithm, and verified the reasonableness of improving a disk brake design problem. The MOMPA, an improvement to the original Marine Predator Algorithm (MPA) in Zhong [13], the method of elite selection is used to improve it, which was tested in CEC2019 and then also on numerous engineering problems; Liu [14] optimized the multi-objective Gray Wolf optimizer(MOGWO) on the perforation parameters to reduce its sacrifice and increase the absorption ratio. Wang [15] inherited the efficient evolutionary mechanism from the Whale Optimization Algorithm (WOA) by combining the Oppositional Learning (OBL) and Global Grid Ranking (GGR) mechanism into the multi-objective WOA, which was tested with benchmark functions and proved the effectiveness of its improvement. Mohammad Reza Pourhassan [16] proposed a multi-objective-based integration model for solving the problem of location, routing, and inventory management of manufacturing centers, distribution centers, and transport vehicles considering the main objectives of cost minimization and maximization of drivers' working time and used two algorithms, MOWOA and NSGA-II, to pair solve this problem. The performance of these algorithms heavily depends on the performance of the original algorithm. If the original algorithm tends to converge prematurely or the quality of the solution set decreases, the improved algorithm may still fail to avoid these issues. New strategies need to be introduced to further enhance the algorithm.

Xue [17] proposed Dung Beetle Optimization Algorithm, a new algorithm for solving single-objective optimization problems, in 2022.DBO changes the position of particles by simulating their behaviors, and the four behaviors of particles, namely rolling, hatching, foraging, and stealing, correspond to different updating strategies. It can effectively explore space and solve complex search and optimization problems in real-world situations.

Given that the DBO algorithm is effective, many practical improvements and applications have been proposed by many scholars. Zhu [18] introduced three strategies to enhance the DBO algorithm. The Good Point Set strategy increases population diversity and helps avoid local optima; dynamic balancing of foraging and spawning behaviors improves the convergence factor, allowing for better global search in the early stages and stronger local exploration in later stages; and an improved spawning region refines the optimal spawning area to explore the solution space more thoroughly.

Additionally, Zhu used a t-distribution variation strategy based on quantum computation to help the algorithm escape local optima faster, testing the improved DBO on problems like trigonometry, diffraction design, and planar sensor coverage, which showed promising results. Huang [19] proposed an improved Dung Beetle Optimization algorithm called Chaotic Mapping DBO-Twin Support Vector Machine (CMDBO- TSVM). The algorithm combines the advantages of TSVM and DBO and improves the diversity of initial solutions by using chaotic mapping to improve the optimization ability. Wang [20] incorporates the idea of quantum state updating into the class inverse learning, which increases the stochasticity of the population. It incorporates Q-learning in the rolling phase to select the best behavioral pattern, which improves search results and performs in the tests of the CEC2017 with good results. Mai C [21] solved the maximum power point tracking problem and the energy loss problem in PV systems based on the DBO algorithm. Li Y [22] optimized the Variational Mode Decomposition (VMD) using an improved Dung Beetle Optimization (DBO) algorithm and constructed a hybrid prediction model based on the Bidirectional Long Short-Term Memory Network with Attention Mechanisms (BiLSTM-A). This approach significantly improved the accuracy and stability of wind speed predictions.

Based on a limited review of related literature, DBO can effectively handle complex optimization problems in various fields. To verify the performance of the DBO algorithm in multi-objective problems and to solve the multi-objective issue of sensor deployment in a 3D area, this study was conducted. Encouraged by these successes, this study proposes a variant of the DBO algorithm called Multi-Objective Dung Beetle Optimization (MODBO) to tackle complex multi-objective optimization challenges. MODBO leverages the strengths of the original DBO, incorporating advanced mechanisms to enhance performance and robustness in solving multi-objective problems. The authors use the test suite of CEC2020 to evaluate the performance of the proposed algorithm. The DBO algorithm utilizes the neighborhood and competitive mechanisms to enhance performance. In addition, since the multi-objective algorithm has more than one solution at the end, people often use Archive to save the solutions obtained after each iteration, and we also use Archive in MODBO to save the candidate solutions, as well as to facilitate the particles in the competition mechanism to find the nearest optimal solution.

According to the above literature, most swarm intelligence algorithms with MOP improvement are overly concerned with enhancing global search ability and accuracy [23]. There is little concern about the concern for the local optimum. Also, a few algorithms mention the search strategy for the globally optimal particles; therefore, the main study of the DBO algorithm in this paper is as follows:

1. To make DBO optimize multi-objective problems, we add non-dominated sorting and external archiving to the DBO algorithm species so that the DBO algorithm can be solved for multi-objective problems.

2. The dung beetle optimization algorithm depends on the global and local optimum. Therefore, we redefine the bootstrapping strategy of the global and local optimum in the multi-objective dung beetle algorithm, which significantly improves the optimization-seeking ability.

3. The MODBO algorithm was tested on 24 kinds of test functions and compared with nine types of standard algorithms to verify the performance of the MODBO.

4. To validate MODBO's ability to solve engineering problems, we proposed and validated a 3D wireless sensor deployment problem.

## 2.  The proposed method

In this section, we describe the DBO algorithm and introduce two enhancement strategies—the competition mechanism and the neighborhood mechanism—that improve the performance of the MODBO algorithm in multi-objective optimization. The DBO algorithm heavily relies on global and local optima during the search process. However, in multi-objective optimization, these optima are not single points but a set of trade-off solutions forming the Pareto front. To guide particles

effectively while maintaining population diversity—which, in this context, refers to the spatial distribution and coverage of non-dominated solutions—we incorporate the competition mechanism into the position update strategy. This mechanism enables particles to adaptively select leaders based on their relative performance, promoting exploration of underrepresented regions of the Pareto front and preventing premature convergence. As a result, the algorithm can maintain a diverse set of solutions and avoid being trapped in local optima when approaching the best approximation of the true Pareto front. Furthermore, the neighborhood mechanism is introduced to enhance robustness and solution uniformity. By allowing each particle to interact primarily with its neighbors rather than the entire population, this mechanism ensures that even if certain regions become less promising, other areas can continue exploring alternative solutions. This distributed search behavior increases the likelihood of discovering the global Pareto front. Detailed descriptions of both mechanisms are provided in Sections 2.2.1 and 2.2.2.

## 2.1. Dung beetle optimization (DBO)

Dung Beetle Optimization (DBO) is a novel algorithm designed to solve single-objective optimization problems. DBO mimics the behaviors of dung beetles to vary the position of particles, using four distinct behaviors: ball rolling, brooding, foraging, and stealing. Each behavior corresponds to a different updating strategy, allowing the algorithm to explore the solution space efficiently. DBO can effectively address complex search and optimization challenges in real-world applications by emulating these natural behaviors.

**2.1.1. Ball-rolling dung beetles.** Dung beetle ball-rolling behavior is defined in obstructed and unobstructed modes. In the unobstructed mode, the dung beetle is cued by celestial cues to move in a specified direction to maintain the particle's straight-line motion. The formula for updating the particle's position during this ball-rolling behavior is given by Eq. 1:

$$x_i^{t+1} = x_i^t + \lambda \cdot k \cdot x_i^{t-1} + b \cdot \left| x_i^t - x_{\text{worst}} \right|$$

(1)

$x_i^t$ is the position of the $i$ individual at t iterations, $\lambda$ simulates a natural factor randomly taken as −1 or 1, $k$ is a random deflection coefficient of $[0, 1]$, $b$ is an arbitrary coefficient, $x_{worst}$ is the position of the worst individual, $\theta \in [0, \pi]$, and dung beetle position will not be updated if $\theta$ is equal to 0, $\frac{\pi}{2}$ or $\pi$.

When the dung beetle is unable to move forward in the face of an obstacle, it needs to change the direction of its movement through the act of dancing; in the algorithm, the behavior of the particle is simulated by the tangent function when the dance is over, it will get a new guide and can continue to roll the ball in the new direction, and the dancing behavior is shown in Equation 2:

$$x_i^{t+1} = x_i^t + tan(\theta) \left| x_i^t - x_i^{t-1} \right|$$

(2)

**2.1.2. Brood balls.** In nature, animals are cautious in their spawning behavior; as shown in Fig 1, dung beetles usually choose a safer place for spawning; inspired by the above discussion, a safe area for simulating dung beetles' spawning using a boundary selection strategy is proposed as in Eq. 3:

$$Lb^* = max \left\{ x_{lbest} \cdot (1 - R), Lb \right\}$$
$$Ub^* = min \left\{ x_{lbest} \cdot (1 + R), Ub \right\}$$

(3)

$x_{lbest}$ is the local optimal solution, $R = 1 - 1/t_{max}$, $t$ is the current number of iterations, $Lb$ is the lower bound, $Ub$ is the upper bound. Dung beetles engage in reproductive behavior after selecting a suitable location. According to Eq. 3, the boundary of the dung beetle is dynamically transformed during the spawning process, which is mainly determined by the value, and the iterative process is expressed in Eq. 4.

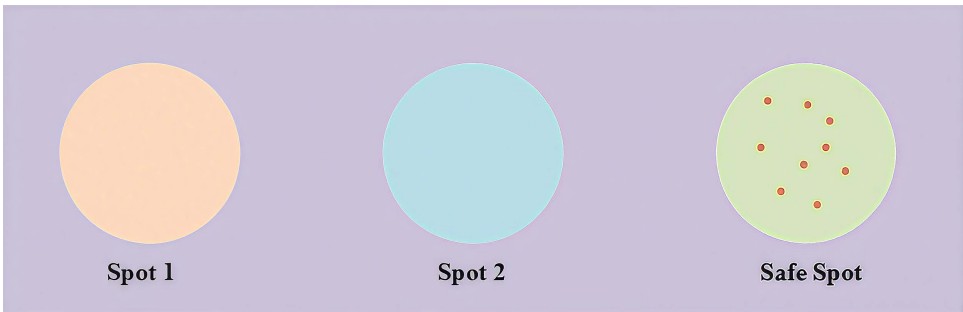

**Fig 1. The boundary selection strategy utilized in the DBO algorithm.**

$$x_i^{t+1} = x_{lbest} + b_1 \times (x_i^t - Lb^*) + b_2 \times (x_i^t - Ub^*) \tag{4}$$

$b_1$, $b_2$ are two D-dimensional independent random vectors.

**2.1.3. Small dung beetle.** Post-birth, juvenile dung beetles exhibit foraging behaviors similar to those of adult dung beetles, foraging exclusively in safe zones. The boundary selection strategy and position updating mechanism employed by juvenile dung beetles are detailed in Equations 5 and 6:

$$\begin{aligned} Lb^* &= max\left\{x_{gbest} \cdot (1-R), Lb\right\} \\ Ub^* &= min\left\{x_{gbest} \cdot (1+R), Ub\right\} \end{aligned} \tag{5}$$

$$x_i^{t+1} = x_i^t + C_1 \times (x_i^t - Lb^*) + C_2 \times (x_i^t - Ub^*) \tag{6}$$

$x_{gbest}$ is the global optimal solution, $C_1$ is a normally distributed random variable, and $C_2$ is a D-dimensional random vector of [0,1].

**2.1.4. Stealing dung beetle.** Stealing Dung Beetles will forage by stealing food from other Dung Beetles, and the best food sources are in the areas where Stealing Dung Beetles are active; Position updated in Eq. 7:

$$x_i^{t+1} = x_{gbect} + S \times g \times \left(\left|x_i^t - x_{gbect}\right| + \left|x_i^t - x_{lbect}\right|\right) \tag{7}$$

$S$ is a constant, and $g$ is a D-dimensional random vector.

Based on Eqs. 1–7, the dung beetle optimization (DBO) algorithm demonstrates strong global search capabilities during the ball-rolling behavior. In another behavior, the algorithm initially conducts a broad global search due to the larger dynamic search range. As the iterations progress, the search range dynamically narrows, leading to a gradual transition toward local search.

From the above discussion, the algorithm flow of DBO is as follows. Firstly, the maximum number of iterations $t_{max}$, and the total number $N$. At the beginning, the DBO algorithm's individuals are initialized. After traversing the various parameters, we get the proportions of the different groups from Fig 2, as shown in Fig 3. The figure uses the fan shape to represent the proportion of particle allocation. In Parts III and IV of this paper, The proportions of each of the four dung beetles were 20%, 20%, 25%, 35%.

**Step 1:** Initialize the algorithm parameters, Population size $N$, Maximum iteration number $t_{max}$, Lower bound $Lb$, Upper bound $Ub$, Dimension $D$, Setting the number of dung beetles with different identities

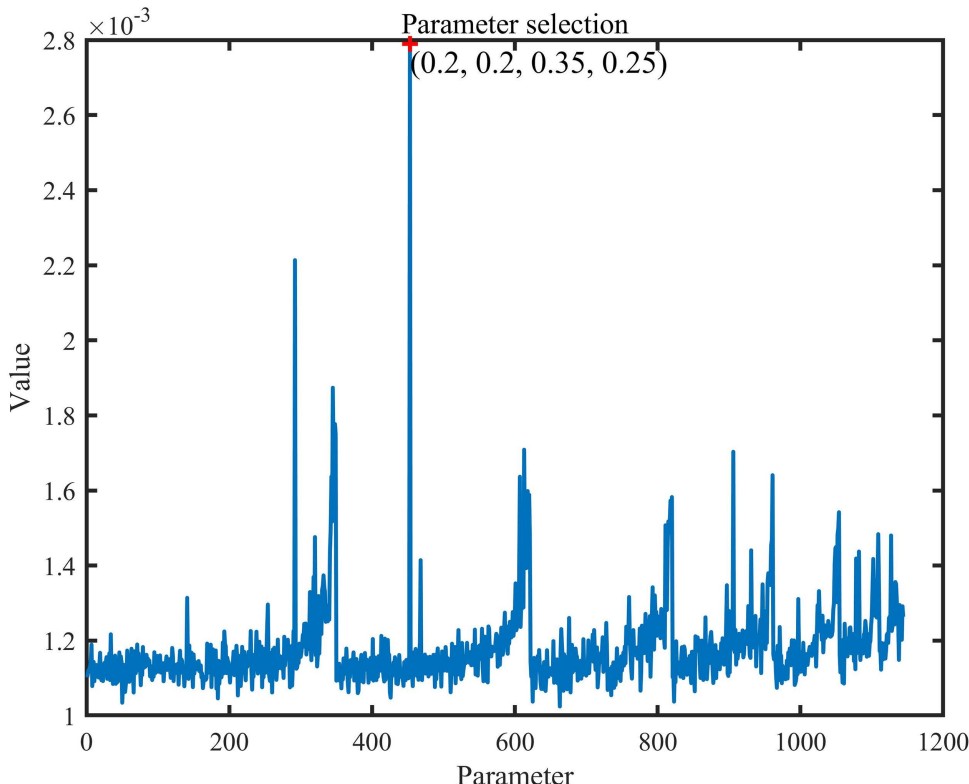

**Fig 2. The hyperparameter selection methodology derived from sensitivity analysis.**

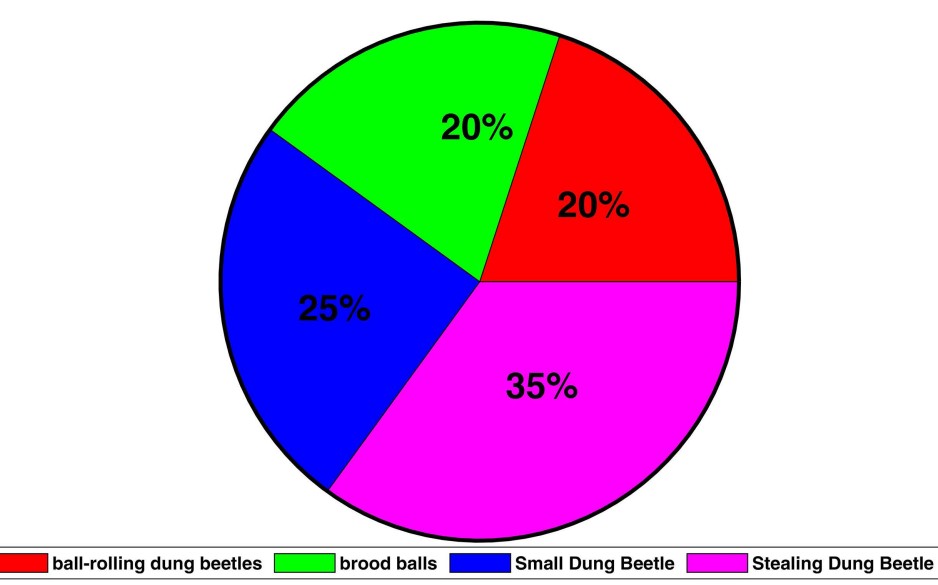

**Fig 3. The distribution strategies of populations within the DBO algorithm.**

**Step 2:** Initialize the population ($X_1, X_2, \ldots, X_{Num}$).

**Step 3:** Obtain the fitness value of the initial population.

**Step 4:** Update the ball-rolling dung beetle position using Eq. 1, 2.

**Step 5:** Calculate the spawning position of the breeding dung beetle using Eq. 3 and update the position of the Brood ball using Eq. 4.

**Step 6:** Update the small dung beetle using Eq. 5 and Eq. 6.

**Step 7:** Update the stealing dung beetle using Eq. 7

**Step 8:** Determine whether individual dung beetles are outside the boundary and update the value of dung beetles outside the boundary using *lb* and *ub*.

**Step 9:** Update the dung beetle's optimal position and best fitness values.

**Step 10:** Repeat the above steps until the termination rule is satisfied and output the global optimal fitness value and its location solution.

## 2.2. The proposed multi-objective dung beetle optimization algorithm

In this section, we introduce the two strategies used to improve MODBO in detail. The overall workflow is illustrated in Algorithm 1.

```
Algorithm 1. General framework of MODBO.
1.   Input: Population size (N_pop), maximum number of iterations (Max_t).
2.   Output: Optimal solution.
3.   Initialize algorithm parameters: Population size (N_pop), maximum number of iterations (Max_t),
     population (p_0)(randomly generated)
4.   For each particle in the population(p_0), calculate its fitness values by evaluating all objec-
     tive functions.
5.   Initialize a Personal Best Archive (PBA) for each particle based on its neighborhood
     information.
6.   Copy the initial population into an archive to keep track of the best solutions found during
     the optimization process.
7.   While (t ≤ Max_t) do
8.   Compute the Personal Best Archive (PBA) for each particle using Algorithm 3.
9.   Update the position of the current particle using the DBO update formula guided by Algorithm 2.
10.  Update the position of each particle using Equations (1)–(7).
11.  Combine P_0 and P_j : P = P_0 ∩ P_j.
12.  Calculate the solutions of the combined P.
13.  Perform non-dominated sorting on P based on NDR and CD.
14.  Replace the original population with the generated P.
15.  t = t + 1
16.  End while
```

**2.2.1. Non-dominated sorting.** Non-dominated sorting is a crucial technique in multi-objective optimization for handling multiple conflicting objectives. It identifies solutions not worse than others in all goals, forming the Pareto front. This technique is essential when converting a single-objective algorithm into a multi-objective one, as it helps manage trade-offs and generate a diverse set of optimal solutions. Non-dominated sorting ensures that algorithms can effectively balance multiple objectives in complex, real-world problems. In a multi-objective optimization problem, we are often faced with finding a set of decision variables $x$, which are required to optimize multiple objective functions $f1(x), f2(x), \ldots, fm(x)$.

 

Here, a solution $x'$ is considered to dominate another solution $x''$ if for all $i \in \{1, 2, \ldots, m\}$ there is $f_i(x') \leq f_i(x'')$ and there exists at least one j such that $f_j(x') \leq f_j(x'')$. In other words, $x'$ is not worse than $x''$

The first level of the non-dominated solution set represents the most significant group of solutions that are not dominated by any other solutions in the current population. These solutions form the Pareto Front, which consists of the best trade-offs among conflicting objectives in the solution space.

After identifying the first level of non-dominated solutions, these solutions are removed from the original population. The same sorting process is then applied to the remaining solutions to determine the second level of non-dominated solutions. This process continues until all solutions are classified to a certain level, thus effectively classifying the solution space hierarchically and helping to identify the near-optimal solution set of the multi-objective optimization problem. This sorting method provides a powerful tool for multi-objective optimization and guides the population's evolutionary direction as part of fitness assessment in evolutionary algorithms.

**2.2.2. Competition mechanism.** In the Dung Beetle Optimization (DBO) algorithm, exploration mainly depends on identifying global and local optima. However, in multi-objective optimization, these optima are not single points but a set of non-dominated solutions forming the Pareto front. To guide particles effectively while maintaining diversity, we propose an enhanced strategy based on a competition mechanism within the population. This mechanism allows less effective particles to learn from better-performing ones, which helps maintain both convergence and diversity. Based on this idea, we introduce a new method for selecting global guiding particles through pairwise random competitions among elite candidates. In each competition, two elites are randomly selected, and the one with the smaller angle relative to the current particle is chosen as the guide. The current particle then updates its position by learning from the winner, as shown in Fig 4. This angle-based selection promotes movement toward promising regions and enhances exploration of underrepresented areas of the Pareto front. The overall framework of the competition mechanism in MODBO is presented in Algorithm 2.

The learning strategy based on the competition mechanism consists of three parts: elite particle selection, pairwise competition, and particle learning. The role of elite particles is used to provide candidate particles for two-pair competition to compete for the best particles to guide the renewal. The process of generating elite particles is as follows: firstly, the

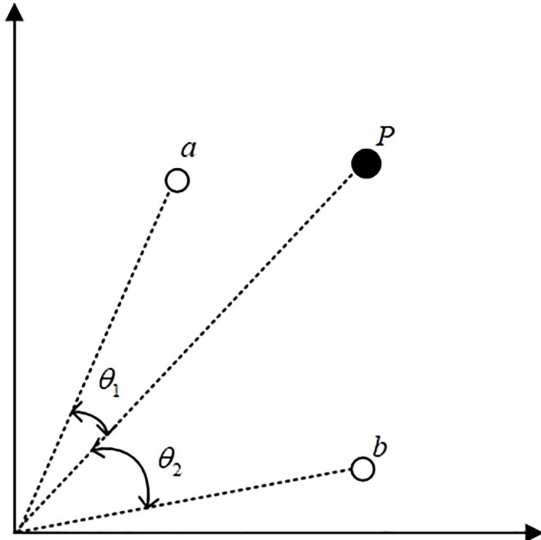

**Fig 4. The particle selection method in competitive mechanisms.**

population is sorted non-dominated to get the Pareto front $F_1$, $F_2$, ..., $F_n$, where $n$ represents the number of all the frontier particles found so far, after which the selection of elite particles is carried out, mainly by the index of the frontier where the particles are located and the congestion distance between them.

The particles compete two by two after being selected from the elite particle set, and the winning elite particle directs the movement of the current particle. In each competition, two particles, a and b, are extracted from the elite particle set, and the angle between particle P and the energetic particles a and b are calculated; the one with the smaller angle wins, and particle P learns in the direction of the smaller angle. Fig 4 illustrates the schematic diagram between the two competitions. When a winner is produced, the particle can learn in the direction of the winner.

**Algorithm 2.** Pseudocode of the Competition Mechanism in MODBO.
```
1.   Perform non-dominated sorting on the population to obtain Pareto fronts F₁,F₂,...,Fₙ
2.   Select elite particles from the first few fronts based on their rank and crowding distance
3.   For i = 1 to length (EliteSet) do
4.     Randomly select two particles a and b from EliteSet
5.     Calculate angle between current particle P and a:θₐ = angle(P − a)
6.     Calculate angle between current particle P and b:θ_b = angle(P − b)
7.     Determine winner w:
8.       if (θₐ<θ_b) then (w = a) else (w = b)
9.     Update particle P's position using w as the guide
10.  Replace P with P_new in the population
11.  Return updated population
```

**2.2.3. Neighborhood mechanism.** In the MODBO, guiding particles using local optima is crucial as it influences the iteration efficiency of DBO. To enhance this aspect, a ring topology was incorporated into the design of MODBO, allowing each particle to interact solely with its immediate predecessor and successor. Each particle ($i$) selects its local optimal guidance based on its Personal Best Archive ($PBA\{i\}$). This approach ensures that particles reference only local information, not the global optimum, thereby improving local search capabilities. The overall framework of the Neighborhood mechanism is presented in Algorithm 3.

Initially, initialize the entire population and set up the PBA for each particle; For particle $i$, the leader is determined through non-dominated sorting, selecting the top-ranked value from PBA{i} as the best option., Using the DBO update formula, adjust the position of particle $i$ from $p_i(t)$ to $p_i(t+1)$. Following the evaluation of the fitness function, store $p_i(t+1)$ into its own PBA{i}, remove the particles dominated by PBA{i}, and put the values of $i-1$ particle, $i$ particle, and $i+1$ particle into PBA{i}, and then sort the four values into a non-dominated sorting. The highest-ranked point serves as the non-dominated particle in the guiding personal best archive, directing the subsequent movement of particle $i$. Repeat the above steps until the termination condition is satisfied.

**Algorithm 3.** Neighborhood mechanism.
```
1.   Initialize population P = {p₁,p₂, ...,pₙ}
2.   Initialize Personal Best Archive (PBA) for each particle i
3.   While termination condition not met do
4.     For each particle i from 1 to n do
5.       Select local optimal guidance from PBA{i} through non-dominated sorting
6.       leader_i=Non_Dominated_Sort(PBA{i})[0]
7.       Pick the top-ranked solution as leader
8.       Update particle position using DBO formula with local guidance
9.       fitness = Evaluate_Fitness(pᵢ(t+1))
10.      Update PBAᵢ with new solution and remove dominated solutions
11.      PBA{i}.append(pᵢ(t+1))
12.      PBA{i} = Remove_Dominated(PBA{i})
13.      Incorporate neighboring particles' information into PBA{i}
```

```
14.      neighbors = [i − 1, i, i + 1]
15.      For each neighbor in neighbors:
16.        If neighbor exists then
17.          Add best solution from neighbor's PBA
18.      Perform non-dominated sorting on updated PBA[i]
19.      sorted_PBA=Non_Dominated_Sort(PBA[i])
20.      Keep only non-dominated solutions
21.      Update local guide to the new best solution
22.    End for
23.  End while
```

## 3. Results and discussion

In this section, we conduct several experiments to assess the performance of the MODBO algorithm in addressing large-scale multi-objective optimization problems (MOPs). We utilize 24 benchmark functions from the CEC2020 suite, which are tailored for evaluating multi-objective optimization algorithms. These functions simulate real-world problem characteristics, including nonlinearity, modality, rotation, translation, and noise, thereby providing a comprehensive evaluation framework.

The CEC2020 test suite comprises 14 bi-objective and 10 tri-objective functions, each varying in dimension, complexity, and features, aimed at testing convergence, diversity preservation, and robustness. To evaluate MODBO's effectiveness, it is compared against nine other algorithms listed in Table 1. Performance is visualized using Pareto Set (PS) and Pareto Front (PF) plots, which illustrate both the distribution and quality of solutions.

### 3.1. Parameter configurations

All experiments were conducted on the same computer to ensure a fair comparison. Considering that the parameter settings of each algorithm can significantly influence its performance, we carefully tuned the parameters of our proposed algorithm through multiple trials to achieve the best possible results. For the other compared algorithms, we first reimplemented them based on the settings reported in their original literature, and then performed manual adjustments to minimize any potential bias caused by parameter configurations. The final parameter settings for all algorithms are summarized in Table 1. The settings of each algorithm are shown in Table 1. The experimental environment used was Windows 11, 64-bit operating system, with an Intel(R) Core(TM) i5-10600KF CPU @ 4.10GHz, and the algorithms were based on the PlatEMO [24] platform.

**Table 1. Configuration settings of the ten comparative algorithms.**

| Algorithm | Settings configuration |
| --- | --- |
| NSGAII [4] | $P_c = 0.8, P_m = 0.1$ |
| MOSCA [12] | $archive_{sizc} = 100, \ a = 2, r_1 = a - \text{CurrentIteration}^*((a)/Maxt), r_2 = 6.28^* rand, r_3 = 2^* rand, r_4 = rand$ |
| MOGWO [25] | $nGrid = 10, \alpha = 0.1, \beta = 10, \gamma = 2$ |
| MOMFO [26] | $\alpha$ is decreased linearly from 2 to 0, $\ b = 1$ |
| MOSSA [27] | $R_2 = 0.8, SD = 0.1, PD = 0.2$ |
| MOGTO [28] | $archive_{size} = 100, p = 0.03, w = 0.8, \beta = 3$ |
| MO_Ring_PSO_SCD [29] | $archive_{size} = 100, C_1 = C_2 = 2.05, W = 0.7298$ |
| SPEA2 [30] | $P_c = 0.8, P_m = 0.1$ |
| MOEA/D [31] | $archive_{size} = 100, T = 0.1, n_r = 0.01, \delta = 0.9, \eta = 30$ |
| MODBO | $K = \ 0.2, b = \ 0.2, \ and \ S \ = 0.25$ |

## 3.2. Performance criteria

This paper uses two performance metrics to evaluate each algorithm's convergence distance and convergence accuracy: Inverted Generational Distance (IGD) and Hypervolume (HV). A smaller IGD value signifies that the solutions generated by the algorithm are closer to the true Pareto front, indicating superior convergence and distribution. Specifically, IGD measures the average distance from the reference Pareto front to the set of solutions found by the algorithm, as defined in Eq. 8. Hypervolume (HV), on the other hand, evaluates the volume of space dominated by the Pareto front obtained by the algorithm. It reflects the size of the region covered by the non-dominated solutions within a specified range. A higher HV score indicates a better distributed set of solutions. The calculation of HV is provided in Eq. 9.

$$IGD(P, Q) = \frac{\sum_{v \in P} d(v, Q)}{|P|}$$

(8)

where $P$ is the true Pareto frontier, $Q$ is the set of true Pareto optimal solutions obtained by DBO, $d(v, Q)$ is the minimum Euclidean distance from individual $v$ in $P$ to $Q$, and $|p|$ is solution sets distributed on the true Pareto surface.

$$HV(Sol) = L\left(U_{\alpha \in Sol} [f_1(a), Ref_1] \times \cdots \times [f_u(a), Ref_u]\right)$$

(9)

Where $L(Z)$ is the Lebesgue metric of $Z$ and $Ref$ represents the sets $(1.2, 1.2)^T$ and $(1.2, 1.2, 1.2)^T$.

## 3.3 Test function

Table 2 shows the properties of the CEC2020 that we use. The proposed test problem is characterized by the presence of local Pareto optimal sets (PSs), a scalable number of PSs, a non-uniformly distributed number of PSs, discrete Pareto fronts (PFs), and a scalable number of variables and objectives.

In this test, the algorithm was run for 20 separate runs and 1000 iterations number for all the test problems to show fairness. The results are placed in Tables 3 and 4. Table 3 shows the mean and SD of the IGD metrics for the ten algorithms in the CEC2020 benchmark test, and Table 4 shows the mean and SD of the HV metrics for the ten algorithms in the CEC2020 benchmark test.

The statistical results of the IGD metrics over the 20 runs of each algorithm are shown in Table 3, and in general, MODBO gets the best results on most algorithms. However, MODBO achieved the second-best performance after NSGA2 on MMF8 and the second-best performance after MO_RING_PSO_SCD on MMF10_I, MMF11_I, MMF12_I, MMF13_I, MMF15_I, MMF15_a_I, MMF16_I1, MMF16_I2, MMF16_I3. The performance is due to the introduction of the ring topology with indexes in MO_RING_PSO_SCD and the use of a unique concept of congestion distance, which possesses excellent results on the three-objective discontinuous multi-peak function. The MEAN results of IGD show that the MODBO algorithm can find values closer to the Pareto-optimal solution, and the SD results of IGD show that MODBO can see the set of Pareto-optimal solutions for the test function more quickly. Similarly, the results of the HV metrics for 20 runs of each algorithm are shown in Table 4. The MODBO algorithm achieves promising results on most of the problems. However, NSGA II outperforms MODBO on MMF8, MMF15_a, and MMF11_I, this is because these three functions have strong deceptive capabilities, especially MMF11_I, which focuses more on testing the local search ability of algorithms and has certain limitations for algorithms with strong global search abilities. Similarly, MOGWO exceeds MODBO on MMF14 and MMF15.on MMF15_I, MO_RING_PSO_SCD outperforms MODBO, this is because MMF15_I contains trap regions, and MO_RING_PSO_SCD performs well in escaping from local optima. Based on the conclusions of the HV and IGD tables, the MODBO algorithm achieves better results on most of the tested problems, and even if it does not achieve the best results, MODBO ranks second or third. In Table 4, the Mean of HV indicates that MODBO has better global search capability with more uniform distribution in the solution space, and this capability helps the algorithm to find more solutions in

**Table 2. Performance indicators of various test functions in CEC2020.**

| MMO test problem name | Scalable number of variables | Scalable number of objectives | Pareto optimal known | Pareto front geometry | Pareto set geometry | Scalable number of ops_local | N_ops |
|---|---|---|---|---|---|---|---|
| MMF1 | × | × | √ | Convex | Nonlinear | × | 2+0 |
| MMF2 | × | × | √ | Convex | Nonlinear | × | 2+0 |
| MMF4 | × | × | √ | Concave | Nonlinear | × | 2+0 |
| MMF5 | × | × | √ | Convex | Nonlinear | × | 2+0 |
| MMF7 | × | × | √ | Convex | Nonlinear | × | 2+0 |
| MMF8 | × | × | √ | Concave | Nonlinear | × | 2+0 |
| MMF10 | × | × | √ | Convex | Linear | × | 1+0 |
| MMF11 | × | × | √ | Convex | Linear | √ | 1+0 |
| MMF12 | × | × | √ | Convex | Linear | √ | 1+0 |
| MMF13 | × | × | √ | Convex | Nonlinear | √ | 1+0 |
| MMF14 | √ | √ | √ | Concave | Linear | √ | 2+0 |
| MMF15 | √ | √ | √ | Concave | Linear | √ | 1+0 |
| MMF1_e | × | × | √ | Convex | Nonlinear | × | 2+0 |
| MMF14_a | √ | √ | √ | Concave | Nonlinear | √ | 2+0 |
| MMF15_a | √ | √ | √ | Concave | Nonlinear | √ | 1+0 |
| MMF10_l | × | × | √ | Convex | Linear | × | 1+1 |
| MMF11_l | × | × | √ | Convex | Linear | √ | 1+1 |
| MMF12_l | × | × | √ | Convex | Linear | √ | 1+1 |
| MMF13_l | × | × | √ | Convex | Nonlinear | √ | 1+1 |
| MMF15_l | √ | √ | √ | Concave | Linear | √ | 1+1 |
| MMF15_a_l | √ | √ | √ | Concave | Nonlinear | √ | 1+1 |
| MMF16_l1 | √ | √ | √ | Concave | Linear | √ | 1+2 |
| MMF16_l2 | √ | √ | √ | Concave | Linear | √ | 2+1 |
| MMF16_l3 | √ | √ | √ | Concave | Linear | √ | 1+2 |

discontinuous functions. The SD values in Table 4 indicate that MODBO has strong stability on most of the tested functions, with more consistent results after several experiments.

Table 5 presents the results of the Wilcoxon rank sum test for the MODBO algorithm compared to other algorithms, indicating the p-value of the Wilcoxon rank sum test at $\propto=$ 5% significance level. If $p < 0.05$, the null hypothesis is rejected in favor of the alternative hypothesis, indicating a significant difference. Table 5 reveals that almost all p-values are below 0.05, demonstrating that MODBO performs significantly better than the other algorithms. Therefore, the MODBO algorithm exhibits remarkable performance in comparison.

Convergence curves are visually analyzed to evaluate the performance of compared algorithms. Fig 5 through 11 present the PS (Population Set) and PF (Pareto Front) plots for MODBO and its competitors. The PS plot reflects the population distribution during the optimization process, while the PF plot illustrates the Pareto front discovered by each algorithm. By comparing these PF plots, one can visually assess the effectiveness of each algorithm. Figs 5-7 show the convergence curves of the comparison algorithm and MODBO on the test function. The functions MMF1, MMF14, and MMF14_a have been chosen as the curve for the display, bi-objective functions, and bi-objective discontinuity functions to validate the PS obtained by the algorithms, and the PS plots of Fig 5 confirm that MODBO generates better PS than the other algorithms. Fig 6 demonstrates that the MODBO algorithm can find more solutions in different spaces. Fig 7 highlights that the MODBO algorithm distributes solutions better than the MO_RING_PSO_SCD algorithm and NSGA II. The MODBO algorithm develops more PSs by utilizing the competitive mechanism.

**Table 3. Obtained IGD values by MODBO and other MO competitors.**

| function | | MO_Ring_PSO_SCD | NSGA2 | MOEA/D | SPEA2 | MOGWO | MOSCA | MOGTO | MOSSA | MOMMFO | MODBO |
|---|---|---|---|---|---|---|---|---|---|---|---|
| MMF1 | Mean | 1.97E-03 | 1.57E-03 | 4.12E-03 | 1.69E-03 | 1.18E-02 | 2.31E-03 | 3.02E-02 | 2.28E-03 | 1.77E-02 | **1.31E-03** |
| | SD | 4.98E-05 | 1.30E-04 | 2.04E-04 | 9.15E-05 | 4.82E-03 | 6.61E-05 | 8.97E-03 | 8.30E-05 | 2.05E-03 | **6.95E-05** |
| MMF2 | Mean | 1.31E-02 | 8.12E-03 | 3.78E-03 | 1.26E-02 | 5.29E-02 | 1.45E-01 | 6.86E-02 | 5.58E-03 | 9.91E-02 | **1.53E-03** |
| | SD | 1.74E-03 | 7.81E-03 | 4.58E-05 | 3.59E-03 | 2.37E-02 | 3.46E-02 | 2.04E-02 | 6.45E-04 | 2.41E-02 | **1.21E-04** |
| MMF4 | Mean | 1.76E-03 | 1.25E-03 | 3.81E-03 | 1.66E-03 | 1.68E-02 | 3.79E-03 | 2.83E-02 | 2.31E-03 | 1.24E-02 | **1.23E-03** |
| | SD | 1.24E-04 | 6.03E-05 | 6.64E-06 | 1.35E-04 | 8.03E-03 | 1.90E-03 | 3.70E-03 | 8.73E-05 | 1.26E-03 | **7.93E-05** |
| MMF5 | Mean | 2.01E-03 | 1.32E-03 | 4.00E-03 | 1.70E-03 | 2.91E-03 | 2.25E-03 | 2.87E-02 | 2.22E-03 | 1.83E-02 | **1.28E-03** |
| | SD | 5.14E-05 | 7.66E-05 | 1.13E-04 | 6.62E-05 | 3.10E-04 | 3.23E-05 | 7.62E-03 | 5.66E-05 | 1.50E-03 | **3.07E-05** |
| MMF7 | Mean | 1.88E-03 | 1.30E-03 | 3.84E-03 | 1.71E-03 | 5.39E-03 | 2.33E-03 | 2.47E-02 | 2.30E-03 | 1.30E-02 | **1.27E-03** |
| | SD | 5.49E-05 | 6.57E-05 | 8.49E-06 | 1.22E-04 | 6.21E-04 | 4.00E-05 | 4.60E-03 | 5.36E-05 | 1.76E-03 | **4.27E-05** |
| MMF8 | Mean | 2.85E-03 | **1.26E-03** | 4.21E-03 | 1.65E-03 | 3.67E-03 | 2.55E-03 | 8.16E-02 | 2.33E-03 | 3.36E-02 | 1.30E-03 |
| | SD | 1.95E-04 | **4.53E-05** | 3.37E-06 | 7.73E-05 | 3.93E-03 | 9.57E-05 | 1.84E-02 | 7.22E-05 | 5.62E-03 | 2.91E-05 |
| MMF10 | Mean | 1.07E-01 | 1.06E-01 | 2.02E-01 | 7.68E-02 | 5.07E-02 | 4.47E-02 | 3.19E-01 | 9.15E-02 | 2.46E-01 | **3.50E-03** |
| | SD | 2.33E-02 | 1.01E-01 | 1.39E-01 | 7.22E-02 | 1.01E-02 | 6.42E-03 | 3.03E-02 | 2.59E-02 | 4.99E-02 | **1.34E-04** |
| MMF11 | Mean | 1.76E-02 | 1.09E-02 | 7.76E-02 | 1.38E-02 | 1.81E-02 | 9.13E-03 | 2.02E-01 | 9.41E-03 | 1.10E-01 | **5.01E-03** |
| | SD | 1.31E-03 | 6.96E-04 | 1.23E-04 | 4.56E-04 | 1.96E-03 | 5.01E-04 | 1.83E-02 | 5.13E-04 | 1.34E-02 | **3.20E-04** |
| MMF12 | Mean | 7.65E-03 | 8.92E-03 | 6.72E-03 | 2.77E-03 | 5.09E-03 | 3.23E-02 | 9.82E-02 | 7.61E-03 | 7.31E-03 | **1.09E-03** |
| | SD | 1.03E-03 | 2.12E-02 | 9.83E-06 | 1.19E-04 | 8.89E-04 | 3.95E-02 | 3.80E-02 | 1.88E-03 | 1.61E-02 | **4.55E-05** |
| MMF13 | Mean | 3.12E-02 | 1.37E-02 | 4.64E-01 | 1.68E-02 | 2.52E-02 | 1.18E-02 | 2.35E-01 | 1.64E-02 | 1.66E-01 | **6.41E-03** |
| | SD | 2.34E-03 | 5.32E-04 | 5.59E-04 | 1.34E-03 | 3.54E-03 | 1.57E-04 | 5.26E-02 | 2.61E-03 | 1.99E-02 | **1.47E-04** |
| MMF14 | Mean | 6.90E-02 | 9.68E-02 | 2.35E-01 | 1.80E-01 | 1.41E-01 | 1.48E-01 | 2.60E-01 | 1.02E-01 | 1.42E-01 | **6.55E-02** |
| | SD | 1.47E-03 | 3.64E-03 | 1.19E-02 | 2.00E-02 | 3.17E-02 | 5.65E-02 | 3.03E-02 | 3.05E-03 | 5.75E-03 | **2.30E-03** |
| MMF15 | Mean | 1.05E-01 | 1.45E-01 | 2.90E-01 | 2.39E-01 | 1.62E-01 | 1.04E-01 | 3.20E-01 | 1.09E-01 | 1.90E-01 | **6.63E-02** |
| | SD | 5.05E-03 | 8.89E-03 | 2.00E-02 | 4.42E-02 | 4.94E-02 | 2.46E-02 | 2.51E-02 | 6.01E-03 | 1.45E-02 | **2.07E-03** |
| MMF1_e | Mean | 7.93E-03 | 8.06E-03 | 5.66E-02 | 1.02E-02 | 1.20E-02 | 3.42E-03 | 8.47E-02 | 3.02E-03 | 1.19E-01 | **1.63E-03** |
| | SD | 1.00E-03 | 1.43E-02 | 1.69E-02 | 5.83E-03 | 4.20E-03 | 4.48E-04 | 3.76E-02 | 1.52E-04 | 2.51E-02 | **3.28E-04** |
| MMF14_a | Mean | 6.76E-02 | 1.00E-01 | 2.35E-01 | 2.07E-01 | 1.67E-01 | 1.85E-01 | 2.53E-01 | 1.04E-01 | 1.48E-01 | **6.65E-02** |
| | SD | 1.37E-03 | 6.50E-03 | 1.94E-02 | 2.61E-02 | 2.27E-02 | 2.08E-02 | 2.43E-02 | 7.35E-03 | 8.08E-03 | **1.55E-03** |
| MMF15_a | Mean | 1.06E-01 | 1.65E-01 | 2.90E-01 | 2.96E-01 | 1.73E-01 | 2.05E-01 | 3.35E-01 | 1.10E-01 | 1.92E-01 | **6.75E-02** |
| | SD | 4.38E-03 | 1.01E-02 | 2.05E-02 | 4.71E-02 | 3.49E-02 | 2.06E-02 | 3.21E-02 | 6.43E-03 | 9.64E-03 | **2.02E-03** |
| MMF10_I | Mean | **1.80E-01** | 1.97E-01 | 2.22E-01 | 2.01E-01 | 2.37E-01 | 2.16E-01 | 2.38E-01 | 2.31E-01 | 1.92E-01 | 1.92E-01 |
| | SD | **1.40E-02** | 1.39E-02 | 1.34E-02 | 1.02E-02 | 1.90E-02 | 6.18E-03 | 2.88E-02 | 9.73E-03 | 3.92E-02 | 1.35E-04 |
| MMF11_I | Mean | **8.25E-02** | 9.14E-02 | 1.69E-01 | 9.41E-02 | 1.02E-01 | 9.31E-02 | 2.30E-01 | 9.30E-02 | 1.62E-01 | 9.07E-02 |
| | SD | **6.38E-03** | 9.81E-05 | 8.08E-05 | 1.19E-03 | 1.81E-03 | 2.64E-04 | 3.01E-02 | 3.05E-04 | 2.04E-02 | 9.74E-05 |
| MMF12_I | Mean | **6.46E-02** | 8.22E-02 | 9.09E-02 | 7.99E-02 | 8.49E-02 | 1.06E-01 | 1.31E-01 | 8.88E-02 | 1.07E-01 | 8.22E-02 |
| | SD | **1.28E-02** | 5.93E-05 | 6.63E-05 | 9.03E-03 | 1.44E-03 | 2.07E-02 | 1.12E-02 | 2.85E-03 | 1.43E-02 | 5.79E-05 |
| MMF13_I | Mean | **9.96E-02** | 1.45E-01 | 6.27E-01 | 1.49E-01 | 1.64E-01 | 1.47E-01 | 2.38E-01 | 1.45E-01 | 2.00E-01 | 1.44E-01 |
| | SD | **2.61E-02** | 9.54E-04 | 3.78E-03 | 4.51E-03 | 8.18E-03 | 8.47E-04 | 5.22E-02 | 8.40E-03 | 4.63E-02 | 4.17E-04 |
| MMF15_I | Mean | **1.65E-01** | 2.03E-01 | 2.74E-01 | 2.61E-01 | 2.43E-01 | 2.30E-01 | 3.29E-01 | 1.90E-01 | 2.27E-01 | 1.78E-01 |
| | SD | **3.28E-03** | 8.61E-03 | 1.35E-02 | 2.26E-02 | 3.11E-02 | 4.57E-02 | 2.51E-02 | 4.55E-03 | 8.69E-03 | 2.55E-03 |
| MMF15_a_I | Mean | **1.67E-01** | 1.97E-01 | 2.78E-01 | 2.86E-01 | 2.36E-01 | 2.51E-01 | 3.46E-01 | 1.96E-01 | 2.31E-01 | 1.79E-01 |
| | SD | **4.08E-03** | 6.56E-03 | 9.45E-03 | 1.36E-02 | 3.92E-02 | 1.91E-02 | 2.39E-02 | 4.07E-03 | 8.62E-03 | 2.07E-03 |
| MMF16_I1 | Mean | **1.30E-01** | 1.64E-01 | 2.82E-01 | 2.45E-01 | 2.08E-01 | 2.25E-01 | 3.35E-01 | 1.65E-01 | 2.22E-01 | 1.47E-01 |
| | SD | **3.22E-03** | 5.93E-03 | 1.90E-02 | 2.77E-02 | 3.57E-02 | 7.59E-02 | 3.82E-02 | 4.45E-03 | 8.63E-03 | 1.75E-03 |
| MMF16_I2 | Mean | **2.02E-01** | 2.43E-01 | 2.74E-01 | 2.89E-01 | 2.78E-01 | 2.52E-01 | 3.57E-01 | 2.27E-01 | 2.50E-01 | 2.31E-01 |
| | SD | **4.92E-03** | 7.51E-03 | 1.41E-02 | 1.28E-02 | 3.05E-02 | 4.02E-02 | 3.11E-02 | 3.82E-03 | 8.37E-03 | 1.69E-03 |
| MMF16_I3 | Mean | **1.63E-01** | 1.98E-01 | 2.78E-01 | 2.65E-01 | 2.32E-01 | 2.52E-01 | 3.46E-01 | 1.96E-01 | 2.37E-01 | 1.89E-01 |
| | SD | **4.26E-03** | 5.08E-03 | 1.61E-02 | 2.31E-02 | 3.17E-02 | 4.82E-02 | 2.96E-02 | 5.20E-03 | 1.44E-02 | 2.77E-03 |
| +/=/- | | **10/0/14** | 1/0/23 | 0/0/24 | 0/0/24 | 0/0/24 | 0/0/24 | 0/0/24 | 0/0/24 | 0/0/24 | 13/0/11 |

**Table 4. Obtained HV values by MODBO and other MO competitors.**

| HV | | MO_Ring_PSO_SCD | NSGA2 | MOEA/D | SPEA2 | MOGWO | MOSCA | MOGTO | MOSSA | MOMMFO | MODBO |
|---|---|---|---|---|---|---|---|---|---|---|---|
| MMF1 | Mean | 8.73E−01 | 8.74E−01 | 8.65E−01 | 8.74E−01 | 8.56E−01 | 8.73E−01 | 8.07E−01 | 8.73E−01 | 8.44E−01 | **8.75E−01** |
| | SD | 1.32E−04 | 4.77E−04 | 7.84E−03 | 2.84E−04 | 8.16E−03 | 1.38E−04 | 2.76E−02 | 7.85E−05 | 4.38E−03 | **3.17E−05** |
| MMF2 | Mean | 8.59E−01 | 8.66E−01 | 8.71E−01 | 8.54E−01 | 8.13E−01 | 6.32E−01 | 5.64E−01 | 8.68E−01 | 6.37E−01 | **8.74E−01** |
| | SD | 2.28E−03 | 7.14E−03 | 5.96E−05 | 8.00E−03 | 2.15E−02 | 6.75E−02 | 2.18E−01 | 1.04E−03 | 6.19E−02 | **2.00E−04** |
| MMF4 | Mean | 5.40E−01 | 5.42E−01 | 5.36E−01 | 5.40E−01 | 5.20E−01 | 5.38E−01 | 4.73E−01 | 5.40E−01 | 5.15E−01 | **5.42E−01** |
| | SD | 3.59E−04 | 6.24E−05 | 5.63E−03 | 4.32E−04 | 1.16E−02 | 3.01E−03 | 2.56E−02 | 3.70E−05 | 3.43E−03 | **6.29E−05** |
| MMF5 | Mean | 8.73E−01 | 8.75E−01 | 8.68E−01 | 8.74E−01 | 8.72E−01 | 8.73E−01 | 8.21E−01 | 8.73E−01 | 8.42E−01 | **8.75E−01** |
| | SD | 1.19E−04 | 7.08E−05 | 4.87E−03 | 2.77E−04 | 3.93E−04 | 1.29E−04 | 2.40E−02 | 1.00E−04 | 5.82E−03 | **5.38E−05** |
| MMF7 | Mean | 8.74E−01 | 8.75E−01 | 8.71E−01 | 8.74E−01 | 8.69E−01 | 8.73E−01 | 8.36E−01 | 8.73E−01 | 8.54E−01 | **8.75E−01** |
| | SD | 1.44E−04 | 1.73E−04 | 4.70E−05 | 2.06E−04 | 9.45E−04 | 7.06E−05 | 4.59E−03 | 1.02E−04 | 2.79E−03 | **4.42E−05** |
| MMF8 | Mean | 4.19E−01 | 4.23E−01 | 4.20E−01 | 4.20E−01 | 4.19E−01 | 4.21E−01 | 2.98E−02 | 4.22E−01 | 3.03E−01 | **4.23E−01** |
| | SD | 9.83E−04 | 2.70E−05 | 2.51E−05 | 1.04E−03 | 5.98E−03 | 3.99E−04 | 2.77E−01 | 1.34E−04 | 6.03E−02 | **1.05E−04** |
| MMF10 | Mean | 1.26E+01 | 1.23E+01 | 1.20E+01 | 1.27E+01 | 1.27E+01 | 1.28E+01 | 1.13E+01 | 1.27E+01 | 1.17E+01 | **1.29E+01** |
| | SD | 8.27E−02 | 3.78E−01 | 7.20E−01 | 2.70E−01 | 3.45E−02 | 2.00E−02 | 8.35E−02 | 5.31E−02 | 1.22E−01 | **1.85E−04** |
| MMF11 | Mean | 1.45E+01 | 1.45E+01 | 1.44E+01 | 1.45E+01 | 1.45E+01 | 1.45E+01 | 1.34E+01 | 1.45E+01 | 1.41E+01 | **1.45E+01** |
| | SD | 4.71E−03 | 1.29E−03 | 3.37E−04 | 3.67E−03 | 7.65E−03 | 5.08E−04 | 1.85E−01 | 8.70E−04 | 8.66E−02 | **1.96E−04** |
| MMF12 | Mean | 1.56E+00 | 1.57E+00 | 1.57E+00 | 1.57E+00 | 1.57E+00 | 1.47E+00 | 1.45E+00 | 1.55E+00 | 1.35E+00 | **1.57E+00** |
| | SD | 9.55E−03 | 1.40E−01 | 1.28E−05 | 6.38E−04 | 8.54E−04 | 1.62E−01 | 5.35E−02 | 1.58E−02 | 1.38E−01 | **1.32E−05** |
| MMF13 | Mean | 1.84E+01 | 1.84E+01 | 1.70E+01 | 1.84E+01 | 1.84E+01 | 1.84E+01 | 1.72E+01 | 1.84E+01 | 1.76E+01 | **1.85E+01** |
| | SD | 1.32E−02 | 9.99E−04 | 3.06E−03 | 5.72E−03 | 1.00E−02 | 7.54E−04 | 1.31E−01 | 1.73E−02 | 1.08E−01 | **3.67E−04** |
| MMF14 | Mean | 2.98E+00 | 2.83E+00 | 2.38E+00 | 2.60E+00 | **3.56E+00** | 3.11E+00 | 2.87E+00 | 2.82E+00 | 2.72E+00 | 2.83E+00 |
| | SD | 2.20E−01 | 6.01E−02 | 3.35E−01 | 4.46E−01 | **6.45E−01** | 9.46E−01 | 3.24E−01 | 7.03E−02 | 3.16E−01 | 4.38E−02 |
| MMF15 | Mean | 4.01E+00 | 4.11E+00 | 3.45E+00 | 3.82E+00 | **5.10E+00** | 4.17E+00 | 3.93E+00 | 4.18E+00 | 3.89E+00 | 4.19E+00 |
| | SD | 1.25E−01 | 5.94E−02 | 2.28E−01 | 3.97E−01 | **6.18E−01** | 2.56E−01 | 3.97E−01 | 7.02E−02 | 2.28E−01 | 6.00E−02 |
| MMF1_e | Mean | 8.53E−01 | 8.67E−01 | −7.49E+01 | 8.52E−01 | 8.56E−01 | 8.68E−01 | −1.20E+00 | 8.72E−01 | −7.30E−01 | **8.74E−01** |
| | SD | 8.84E−03 | 1.11E−02 | 2.27E+01 | 1.91E−02 | 6.90E−03 | 5.81E−03 | 2.25E+00 | 4.49E−04 | 1.04E+00 | **6.95E−04** |
| MMF14_a | Mean | 2.89E+00 | 2.86E+00 | 2.47E+00 | 2.60E+00 | **3.36E+00** | 3.15E+00 | 2.79E+00 | 2.86E+00 | 2.83E+00 | 2.88E+00 |
| | SD | 1.84E−01 | 5.47E−02 | 5.01E−01 | 4.52E−01 | **5.81E−01** | 2.54E−01 | 3.04E−01 | 6.45E−02 | 4.09E−01 | 3.97E−02 |
| MMF15_a | Mean | 4.08E+00 | **4.61E+00** | 3.56E+00 | 3.59E+00 | 4.18E+00 | 4.55E+00 | 3.57E+00 | 4.30E+00 | 3.92E+00 | 4.41E+00 |
| | SD | 8.74E−02 | **1.43E−01** | 3.24E−01 | 4.31E−01 | 7.15E−01 | 6.28E−01 | 2.64E−01 | 5.93E−02 | 3.25E−01 | 4.21E−02 |
| MMF10_l | Mean | 1.27E+01 | 1.27E+01 | 1.22E+01 | 1.29E+01 | 1.27E+01 | 1.28E+01 | 1.13E+01 | 1.27E+01 | 1.16E+01 | **1.29E+01** |
| | SD | 2.19E−02 | 1.99E−01 | 7.17E−01 | 2.87E−02 | 3.98E−02 | 2.07E−02 | 8.08E−02 | 4.44E−02 | 1.05E−01 | **1.26E−04** |
| MMF11_l | Mean | 1.45E+01 | **1.45E+01** | 1.44E+01 | 1.45E+01 | 1.45E+01 | 1.45E+01 | 1.36E+01 | 1.45E+01 | 1.41E+01 | 1.45E+01 |
| | SD | 1.55E−03 | **4.11E−04** | 2.18E−04 | 2.68E−03 | 4.43E−03 | 4.98E−04 | 1.27E−01 | 5.94E−04 | 7.44E−02 | 3.90E−04 |
| MMF12_l | Mean | 1.57E+00 | 1.57E+00 | 1.57E+00 | 1.57E+00 | 1.57E+00 | 1.48E+00 | 1.45E+00 | 1.56E+00 | 1.41E+00 | **1.57E+00** |
| | SD | 3.79E−03 | 4.67E−05 | 1.07E−05 | 5.18E−04 | 6.14E−04 | 1.30E−01 | 5.57E−02 | 6.51E−03 | 7.26E−02 | **1.97E−05** |
| MMF13_l | Mean | 1.84E+01 | 1.85E+01 | 1.68E+01 | 1.84E+01 | 1.84E+01 | 1.84E+01 | 1.71E+01 | 1.84E+01 | 1.77E+01 | **1.85E+01** |
| | SD | 7.66E−03 | 2.39E−04 | 5.95E−01 | 2.33E−03 | 1.13E−02 | 7.11E−04 | 1.24E−01 | 2.75E−02 | 8.62E−02 | **1.87E−04** |
| MMF15_l | Mean | **4.93E+00** | 4.21E+00 | 3.58E+00 | 3.99E+00 | 4.27E+00 | 4.52E+00 | 4.08E+00 | 4.24E+00 | 3.87E+00 | 4.21E+00 |
| | SD | **1.82E−01** | 8.79E−02 | 3.79E−01 | 2.81E−01 | 5.27E−01 | 9.64E−01 | 4.17E−01 | 1.05E−01 | 4.54E−01 | 5.42E−02 |
| MMF15_a_l | Mean | 4.26E+00 | 4.26E+00 | 3.51E+00 | 3.73E+00 | 4.27E+00 | 4.43E+00 | 3.54E+00 | 4.26E+00 | 3.76E+00 | **4.57E+00** |
| | SD | 8.45E−02 | 7.32E−02 | 3.59E−01 | 3.41E−01 | 5.54E−01 | 4.05E−01 | 3.73E−01 | 7.63E−02 | 2.68E−01 | **5.45E−02** |
| MMF16_l1 | Mean | 4.28E+00 | 4.24E+00 | 3.74E+00 | 3.89E+00 | 4.20E+00 | 3.80E+00 | 4.23E+00 | **4.55E+00** | 3.79E+00 | 4.27E+00 |
| | SD | 3.56E−01 | 4.53E−02 | 3.00E−01 | 4.09E−01 | 6.83E−01 | 5.69E−01 | 3.57E−01 | **1.05E−01** | 3.25E−01 | 6.61E−02 |
| MMF16_l2 | Mean | 4.15E+00 | 4.26E+00 | 3.58E+00 | 4.00E+00 | **4.74E+00** | 4.31E+00 | 3.96E+00 | 4.23E+00 | 3.96E+00 | 4.23E+00 |
| | SD | 2.29E−01 | 5.18E−02 | 2.12E−01 | 2.49E−01 | **4.46E−01** | 6.36E−01 | 6.16E−01 | 1.05E−01 | 2.42E−01 | 4.43E−02 |
| MMF16_l3 | Mean | 4.43E+00 | 4.24E+00 | 3.54E+00 | 3.82E+00 | 4.23E+00 | 4.48E+00 | 3.89E+00 | 4.21E+00 | 3.87E+00 | **4.80E+00** |
| | SD | 1.19E−01 | 2.63E−02 | 4.82E−01 | 2.84E−01 | 4.50E−01 | 1.05E+00 | 4.21E−01 | 9.34E−02 | 3.54E−01 | **4.31E−02** |
| +/=/− | | 1/0/23 | 2/14/7 | 0/0/24 | 0/0/24 | 4/0/20 | 0/0/24 | 0/0/24 | 1/0/24 | 0/0/24 | **16/0/9** |

**Table 5. Wilcoxon test for MODBO and other MO competitors.**

| Test metric | MODBO vs. MO_Ring_PSO_SCD | MODBO vs. NSGA2 | MODBO vs. MOEA/D | MODBO vs. SPEA2 | MODBO vs. MOGWO | MODBO vs. MOSCA | MODBO vs. MOGTO | MODBO vs. MOSSA | MODBO vs. MOMMFO |
|---|---|---|---|---|---|---|---|---|---|
| IGD | 2.33e-03 | 4.29e-05 | 1.82e-05 | 4.39e-05 | 1.82e-05 | 1.88e-05 | 1.88e-05 | 5.60e-05 | 2.70e-05 |
| HV | 2.6958e-05 | 2.69e-05 | 2.41e-03 | 8.80e-03 | 2.69e-05 | 3.08e-05 | 2.61e-04 | 2.69e-05 | 5.52e-05 |

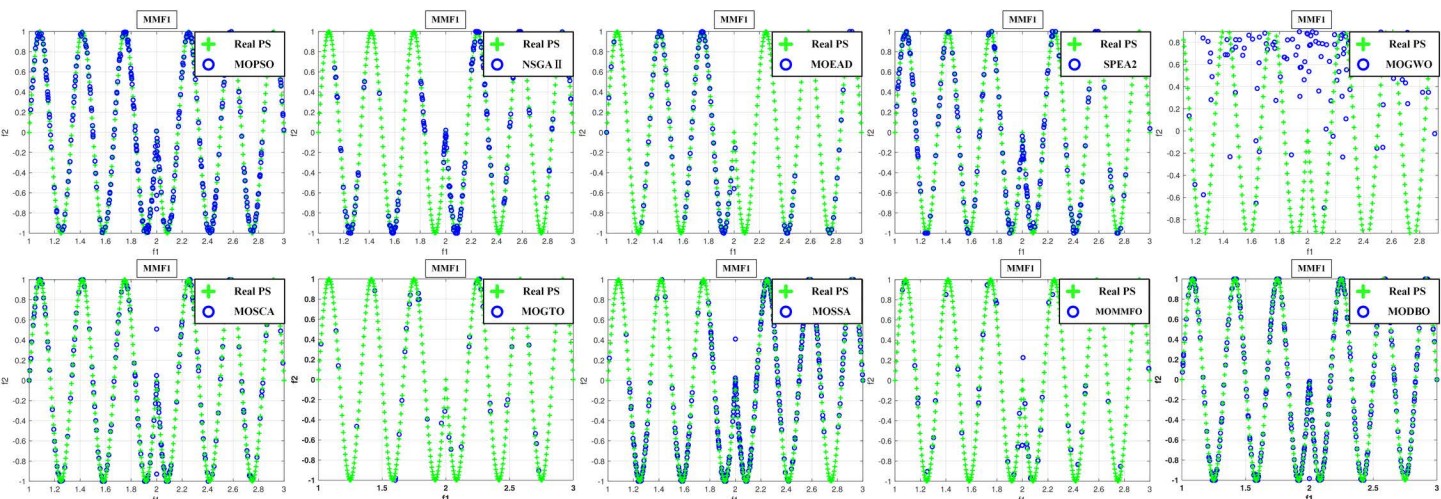

**Fig 5. The PS of ten comparative algorithms on the MMF1 test function.**

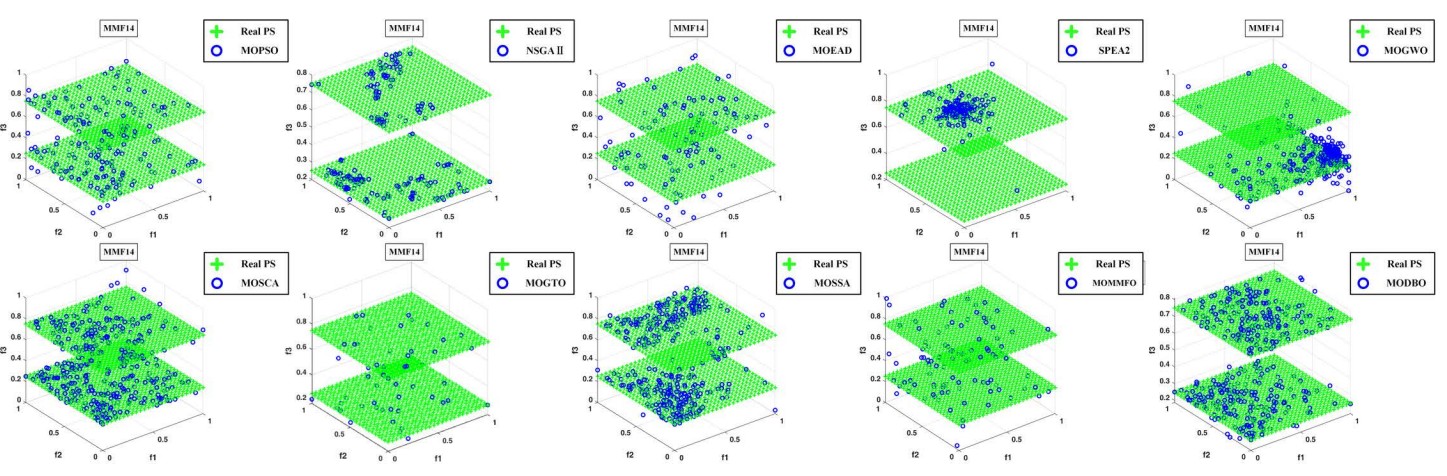

**Fig 6. The PS of ten comparative algorithms on the MMF14 test function.**

Fig 8 to 10 present PF curves for three representative test functions. Fig 8 demonstrates that MODBO has a higher PF fitness than the contrast algorithms on the bi-objective test functions. Fig 9 highlights MODBO is also better at finding solutions in more forms on discontinuous functions. Fig 10 shows that MODBO outperforms other algorithms on triple-objective test functions with a uniform distribution of solutions.

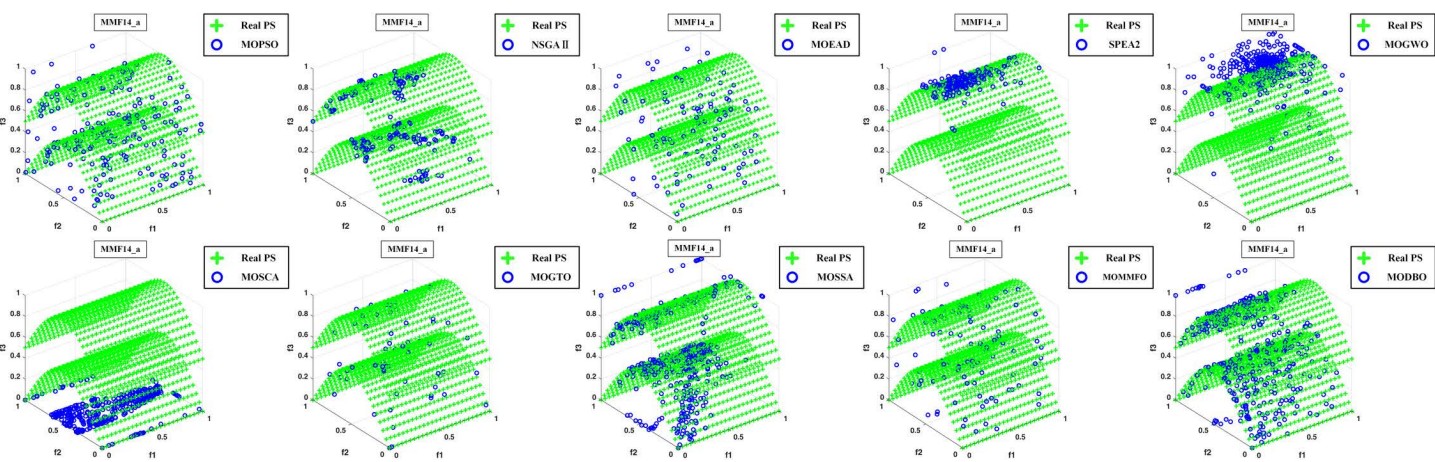

**Fig 7. The PS of ten comparative algorithms on the MMF14a Test Function.**

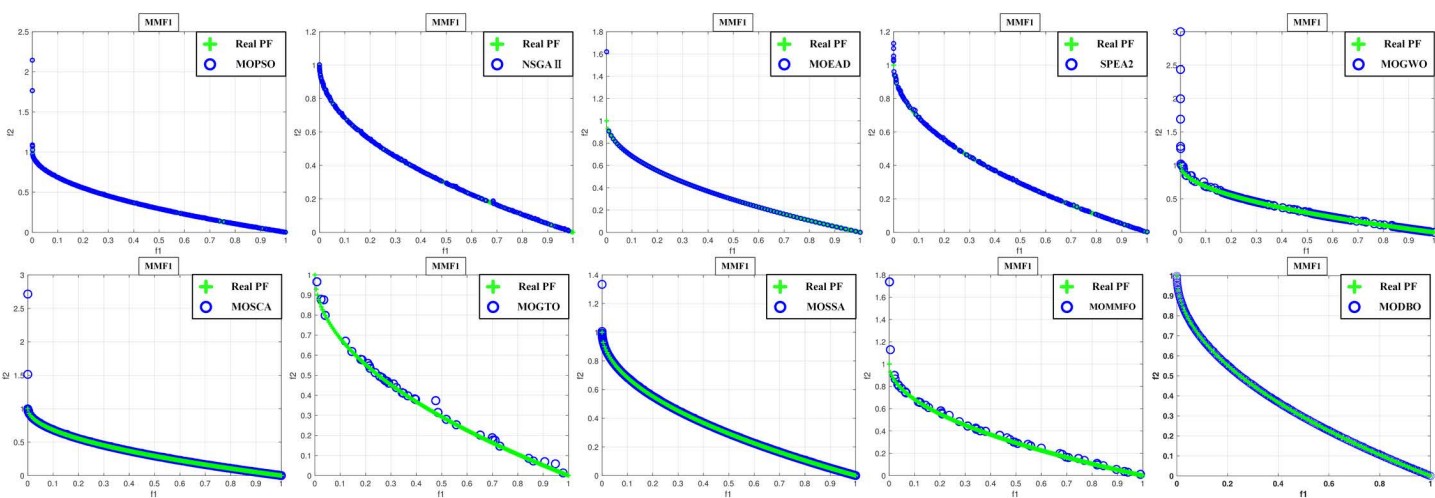

**Fig 8. The PF of ten comparative algorithms on the MMF1 test function.**

Figs 5-10 show that MODBO can provide competitive results on multi-objective algorithms. The IGD statistics reveal MODBO's convergence ability. The high convergence of MODBO comes from the competitive mechanism introduced in the algorithm, which can make the particles guide the position update according to the value closer to them. The HV values and convergence curve graphs illustrate MODBO's solution distribution performance. This distribution capability arises from the neighboring mechanism, where particles update their positions based on the optimal values of their adjacent particles. This approach strengthens the algorithm's exploration ability, allowing it to find solutions more effectively. Overall, the experimental results indicate that MODBO exhibits superior performance on both bi-objective and tri-objective functions, demonstrating its robustness and versatility in solving complex optimization problems.

## 4. Wireless sensor deployment problems

To demonstrate the practical effectiveness of MODBO in real-world scenarios, we apply it to the 3D sensor deployment problem — a complex and challenging optimization task with significant implications in wireless sensor networks (WSNs).

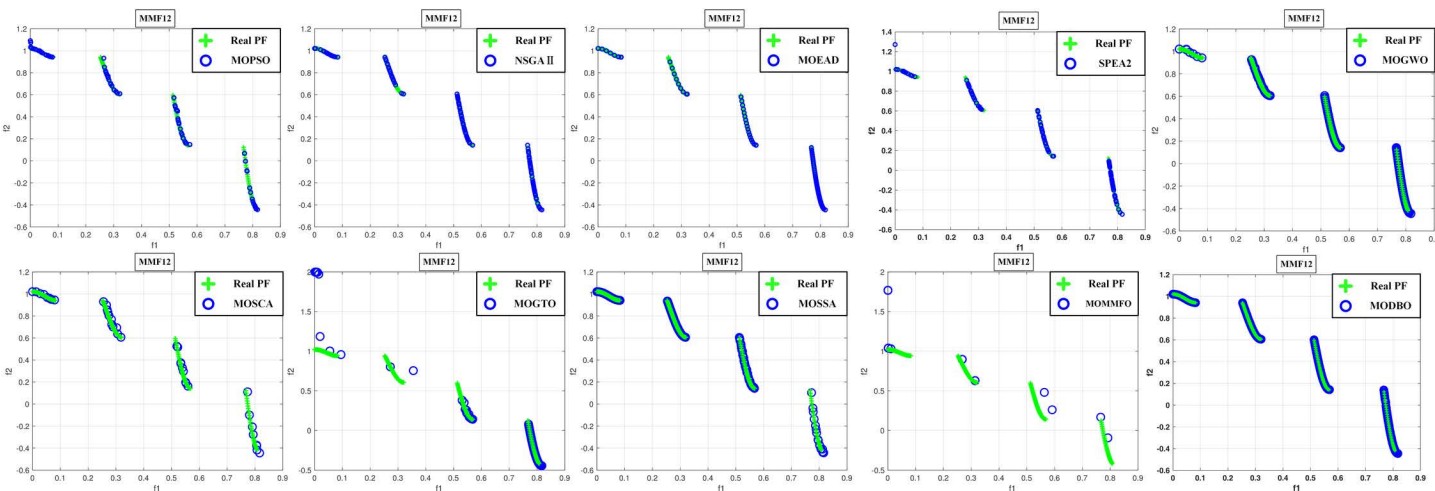

**Fig 9. The PF of ten comparative algorithms on the MMF12 test function.**

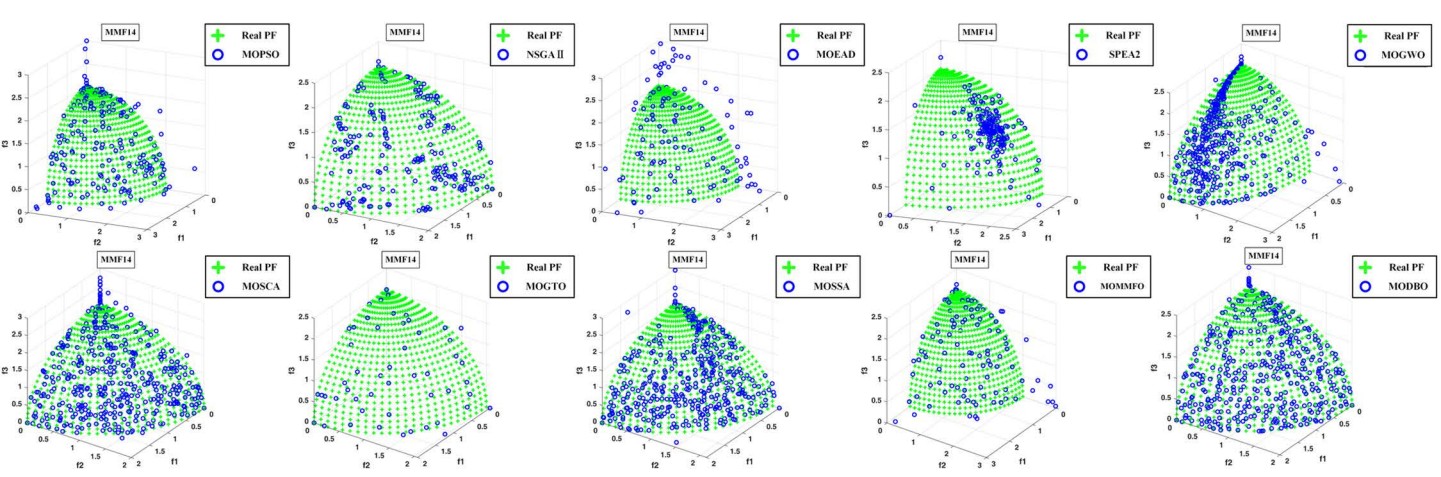

**Fig 10. The PF of ten comparative algorithms on the MMF14 test function.**

This problem involves determining the optimal spatial distribution of sensors in a three-dimensional environment to simultaneously maximize coverage and minimize deployment cost.

The 3D sensor deployment problem is particularly suitable for evaluating MODBO due to several reasons. First, it is inherently multi-objective, requiring a trade-off between two conflicting objectives: maximizing sensing coverage and minimizing the number or cost of deployed sensors. Second, the search space is highly complex due to the 3D spatial arrangement and overlapping sensing ranges, which pose challenges for traditional optimization algorithms in maintaining both convergence and diversity. Third, this problem often exhibits multiple local optima, making global search capability and exploration–exploitation balance crucial — features that MODBO is specifically designed to address.

By applying MODBO to this realistic and challenging scenario, we aim to validate not only its theoretical performance on benchmark functions but also its potential for solving practical engineering problems with real-world constraints.

In this section, the MODBO algorithm will be used to verify its effectiveness in complex engineering problems, and we apply the proposed algorithm to the sensor deployment problem. Swarm intelligence algorithms have had

numerous applications in sensor problems, and Konstantinidis [32] proposed a decomposition-based multi-objective evolutionary algorithm (MOEA/D) and a problem-specific generalized subproblem-related heuristic (GSH) to address the dense deployment and power allocation problem (d-DPAP) in wireless sensor networks (WSNs). The d-DPAP is decomposed into scalar subproblems for parallel optimization. xiong [33] considered the network lifetime and coverage of WSNs to repair the coverage holes using a multi-objective particle swarm optimization algorithm with an adaptive mesh to repair coverage holes. They also employed an improved binary multi-objective evolutionary algorithm based on non-dominated sorting and bi-directional local search to schedule sensor nodes into non-disjoint subsets, extending the network lifetime. Shally Gupta [34] proposed an enhanced deep reinforcement learning algorithm based on dynamic load balancing to improve load balancing performance in fog computing in IoT environments. The algorithm combines Grey Wolf optimization (GWO) and modified Moth Flame algorithm (MFO) and performs well in terms of throughput, latency, response time, and energy consumption. It is also more efficient than existing techniques. In contrast, this paper proposes a 3D sensor deployment method that can be deployed in blind areas in 3D scenes and validated in complex scenes. To confirm the high adaptability of the MODBO algorithm compared to other algorithms, we compare the MODBO algorithm with MOEAD, MOSCA and MOSSA algorithms using the same parameters as in the previous section.

## 4.1. Coverage description

The set of sensor nodes can be denoted as $P = \{p_1, p_2, p_3, ..., p_n\}$, the coverage radius of the sensor nodes is 1, the set of monitoring nodes can be denoted as $Q = \{q_1, q_2, q_3, ..., q_n\}$, the coordinates of the sensor nodes and the monitoring nodes can be represented as $(x_i, y_i, z_i)$ and $(x_j, y_j, z_j)$. the Euclidean distance between the sensor and the node can be denoted as:

$$distence(p_i, q_j) = \sqrt{(x_i - x_j)^2 + (y_i - y_j)^2 + (z_i - z_j)^2}$$

(10)

The probability of being perceived by the monitoring node is:

$$\rho(p_i, q_j) = \begin{cases} 1 & if\ distence(p_i, q_j) < r \\ 0 & otherwise \end{cases}$$

(11)

The joint perception probability of all sensor nodes to the monitoring node is:

$$C(p_{all}, q_j) = 1 - \prod_{i=1}^{n}(1 - \rho(p_i, q_j))$$

(12)

Where $p_{all}$ represents all the sensor nodes in the range, we define L and W as the length and width when mapped to a 2D area. If the value of Eq. 12 is 1, it means that all nodes in the range of the monitoring area where the node is located are covered by node $j$. The area of the sensor coverage area is the sum of the areas of the small areas of all protected nodes, and the total coverage is the sum of the coverage areas of all sensor nodes.

## 4.2. Determination of 3D perception blind zone

As shown in Fig 11, points A, B, and C are all perceptible points in the perception range of point S. However, obstacles between them impede the direct perception of these nodes. As a result, only point A can be perceived, and the two points BC belong to the perception blind zone. To accurately identify such blind spots and improve the reliability of coverage analysis, we propose Algorithm 4.

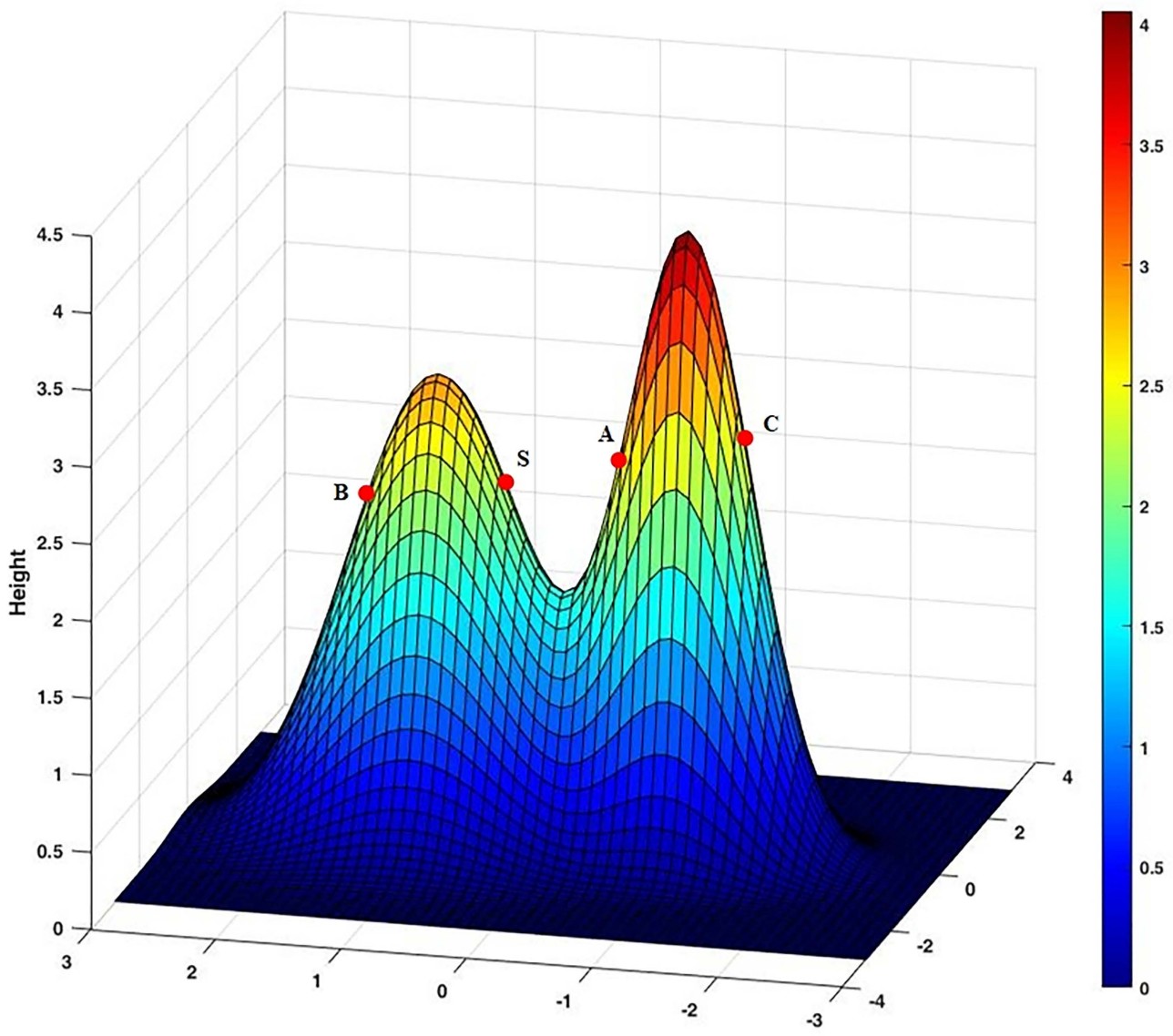

**Fig 11. The schematic illustration of a 3D perception blind zone.**

**Algorithm 4.** Perceptual blind detection.
```
1.   Input: Sensor positions S = {s′, s′, …, s′}, Terrain model T
2.   Output: Set of blind spots B
3.   Initialize B as an empty set
4.   For each target point p in T:
5.     visible=false
6.     For each sensor s in S:
7.       If p is within sensing range of s:
8.         If line-of-sight from s to p is unobstructed (using Eq. 13):
9.           visible=true
10.           Break
11.         Else:
```

```
12.          Check midpoint of s−p using Eq. 14
13.           If midpoint is above terrain surface:
14.            visible=true
15.            Break
16.   If visible==false:
17.      Add p to B
18.  Return B
```

We can use Eq. 13 to determine whether point C can be perceived, assuming that the coordinates of S and C are $(x_1, y_1\_, z_1)$, $(x_2, y_2, z_2)$, and connecting the two points generates the line SC, which can be obtained by combining the two coordinates with the surface equation:

$$\begin{cases} \frac{x-x_1}{x_2-x_1} = \frac{y-y_1}{y_2-y_1} = \frac{z-z_1}{z_2-z_1} \\ \qquad z = f(x, y) \end{cases} \tag{13}$$

If Eq. 13 has no solution, there is no covering point between the two points, and if the solution to Eq. 13 is greater than or equal to 1, there is an intersection of line SC with the curve, and point C is not covered, and the point C fails to be covered.

In addition, there is another case: point B, the line SB, and the curve are not covered, but point B still cannot be detected, so you need to find the coordinates of the midpoint of SB to determine it.

$$\rho(S, B) = \begin{cases} 1 & if \; f(x_3, y_3) \geq z_3 \\ 0 & otherwise \end{cases} \tag{14}$$

To summarize, to determine whether a sensor node can sense a monitoring node, three conditions need to be satisfied, i.e., the value of Eq. 12 is 1, Eq. 13 has no solution, and the value of Eq. 14 is 1.

### 4.3. Problem definition

When sensor nodes are deployed on complex terrain, the more extensive their coverage, the better the deployment effect. However, in practical applications, deployment cost — particularly the number of sensors used — must also be taken into account. In this study, we focus on two primary objectives: maximizing coverage and minimizing the number of deployed sensors. This is because, under static deployment settings, these two goals form the foundation of effective coverage optimization and directly reflect both performance and economic efficiency. While more comprehensive factors such as energy consumption or network lifetime are important, they are often considered in dynamic operational phases and are beyond the scope of this work. Therefore, while ensuring maximum coverage, it is also essential to minimize deployment costs, i.e., to use the fewest possible sensors. Based on this formulation, the sensor deployment problem can be described by the following equation, where N represents the *Nth* deployment scenario:.

$f1(N)$ is the maximum coverage rate:

$$f_1(N) = Max(C(p_{all}, q_j)) \tag{15}$$

$f2(N)$ is the minimum node

$$f_2(N) = Min(Num(n)) \tag{16}$$

## 4.4. Discussion of results

To verify the performance of the MODBO algorithm in real applications, we designed a complex terrain to test the MODBO algorithm. In this experiment, the sensing radius of each sensor was set to 1 meter, the 2D of the surface is a square area with a length and width of 10 meters, the total area of the surface is 122.07 square meters, and the maximum coverage area of the nodes is $S = \pi r^2$=3.14 square meters. Ideally, only 39 nodes are needed to cover it entirely; however, considering the effects of the 3D blind spot, the sensors needed in practice are much larger than the ideal situation.

To evaluate the performance of sensor deployment more rigorously, we conducted 50 independent experimental runs under identical conditions to obtain statistically meaningful results. In each run, 100 sensors were randomly deployed, and the resulting coverage was recorded. Across all trials, the average coverage rate was 84.73%±2.34% (mean±standard deviation), with a representative example shown in Fig 12, where red dots indicate sensor locations and black dots represent monitored areas. The best-performing random trial achieved a coverage rate of 85.91%.

Fig 13 illustrates the optimized deployment using the MODBO algorithm. As shown in the figure, MODBO achieved a significantly higher coverage rate of 93.71% using only 42 nodes, representing an improvement of 7.8% over random placement. The corresponding coverage area is 114.39 (in normalized units). Notably, in the 2D visualization, certain areas display unique color patterns that correspond to elevated regions in the 3D terrain. These high-altitude zones were effectively targeted by the MODBO algorithm for sensor placement, demonstrating its ability to adapt to complex topographical features. This result indicates that MODBO not only enhances overall coverage but also intelligently optimizes node distribution in response to terrain variations, outperforming traditional random deployment methods in terms of both efficiency and effectiveness. The coverage rates of competing algorithms are illustrated in Fig 14. The comparison shows that the MODBO algorithm can accurately find the coverage method with the largest coverage rate and the least number of nodes used. In contrast, other algorithms fail to achieve comparable results within the same number of iterations. This demonstrates that the enhanced MODBO algorithm is particularly effective for deploying sensor networks on complex terrains. An important observation is that the MODBO algorithm avoids falling into local optima on complex surfaces. This is because the MODBO algorithm introduces a competition mechanism, not through a single particle to guide the evolution of the population, but for different particles have different guidance mechanisms, which makes it difficult for the MODBO algorithm to fall into the local optimum, in the later iterations, also through the introduction of the neighborhood mechanism, so that the MODBO's every search is not always a blind search, which improves its accuracy. In conclusion,

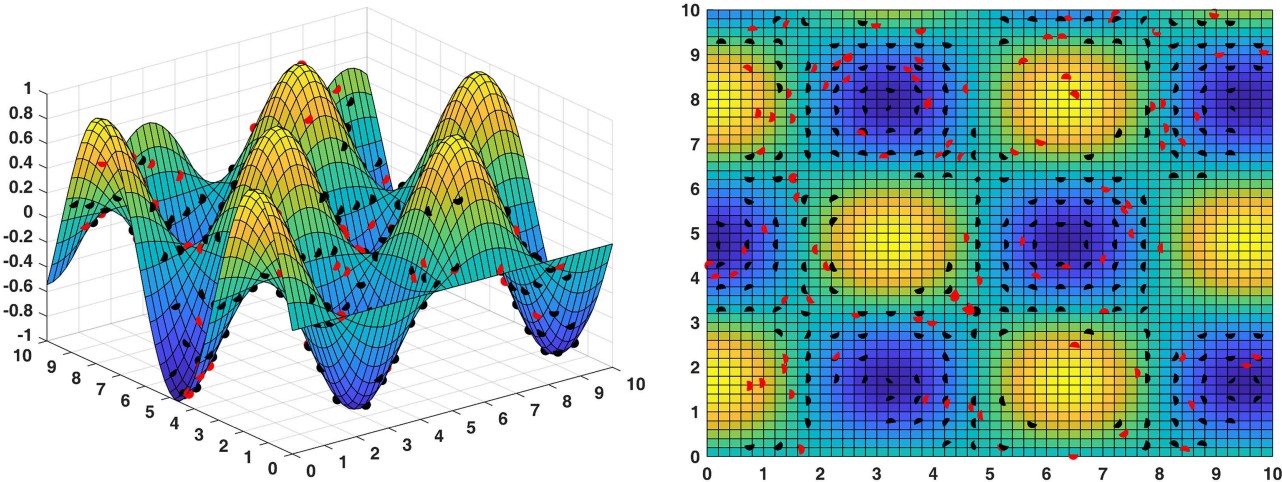

**Fig 12. The sensor coverage map generated by random sensor deployment.**

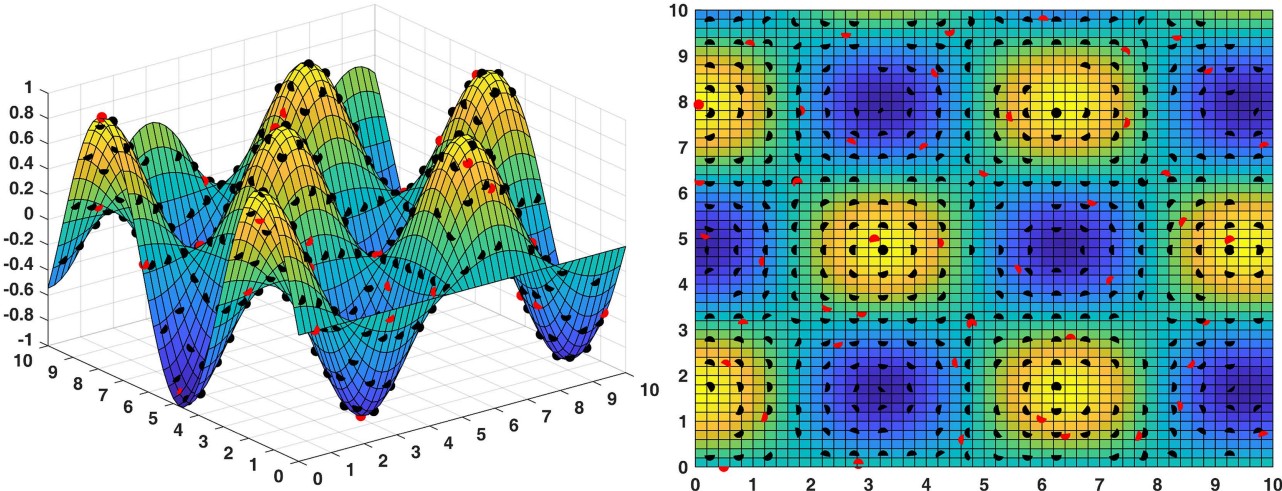

**Fig 13. The optimized sensor coverage using the MODBO algorithm.**

MODBO improves the ability of local and global search, which leads to higher search accuracy and better search results, making it possible to obtain maximum coverage while using the minimum number of nodes when deployed in wireless sensor networks.

### 4.5. Limitation

We need to explore the limitations of our article from two perspectives. At the algorithmic level, we implemented two improvement strategies, which significantly enhanced the algorithm's capability in handling multi-objective problems. However, there are still some limitations. The performance of the algorithm on the bi-objective problem is more than on the tri-objective problem. The primary reason is in the selection of the non-dominated sorting algorithm. As the number of objectives increases, the effectiveness of the non-dominated sorting will gradually diminish when there are more than four objectives. The non-dominated sorting approach of this article will be ineffective, the solution to this problem lies in the proposal of a more effective non-dominated sorting approach to solving the high-dimensional multi-objective optimization problems. Secondly, Secondly, regarding dynamic environments, the two proposed improvement strategies fail to detect dynamic changes in the solution set. This shortcoming can easily cause the algorithm to lose its searchability and become ineffective in handling dynamic problems. A potential solution involves incorporating a dynamic environment detection strategy, enabling the algorithm to adapt and perform well even when faced with dynamic changes. Alternatively, we could integrate reinforcement learning techniques into the algorithm. By doing so, the algorithm would be capable of learning from its interactions with more complex environments, thereby improving its performance in highly dynamic and unpredictable scenarios.

In this study, the coverage was modeled purely from a geometric perspective, focusing on spatial deployment and sensing range, without incorporating other critical aspects of real-world WSN deployments such as communication constraints, power limitations, or data routing mechanisms. While our current formulation assumes a relatively stable deployment environment, we acknowledge that practical sensor network applications often involve more complex and dynamic conditions. Specifically, the current optimization framework does not consider common constraints found in actual WSN deployments — such as maximum allowable energy consumption per node, deployment cost limits, or latency requirements. Additionally, visibility detection is based on simplified geometric models rather than more accurate methods like ray tracing or voxel-based visibility analysis, which could better capture occlusions and blind spots in complex 3D terrains.

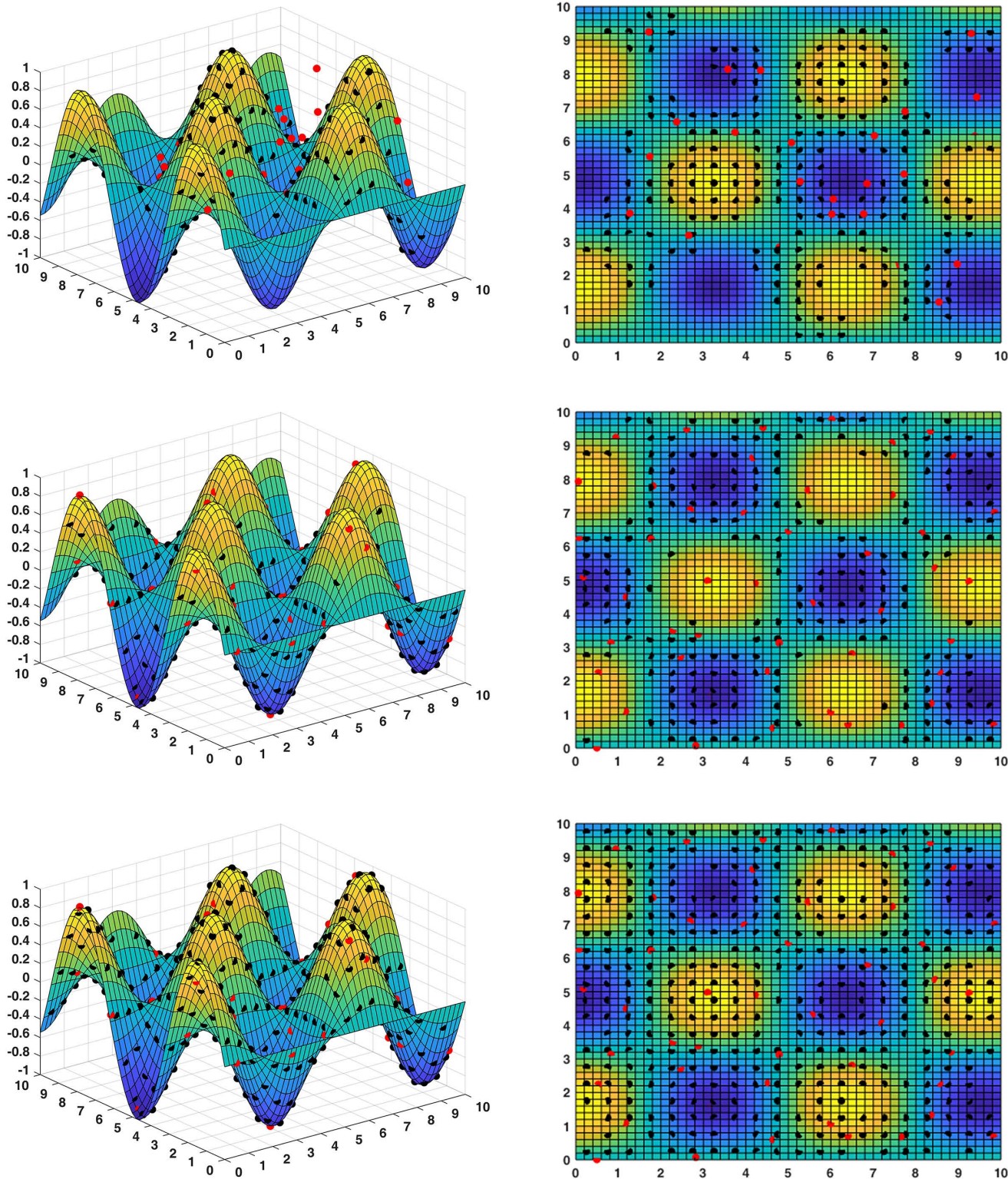

**Fig 14. The comparison of sensor deployment among three algorithms.**

Furthermore, our model assumes ideal sensor behavior and does not account for sensor failures or environmental changes that may affect performance. For instance, natural obstacles or terrain variations might lead to unexpected shifts in sensor positions, which the current algorithm cannot detect or adapt to in real time. Similarly, certain terrain types — such as lakes, marshes, or inaccessible areas — impose physical limitations on where sensors can be placed, but these are not considered in the current setup.

Lastly, the model does not incorporate redundancy mechanisms to ensure continued coverage in the event of sensor failure. In real-world deployments, it is essential to design partial redundancy into the system so that when one sensor fails, others can still maintain coverage over its area. To address these limitations in future work, we propose several directions for model extension: Integration of Communication Constraints: Extend the model to include communication range and data transmission paths, ensuring that coverage also implies connectivity. Incorporation of Energy Models: Introduce energy-aware deployment strategies by modeling power consumption and sensor lifetime, enabling more realistic simulations of long-term network operations. Adoption of Advanced Visibility Models: Replace the current geometric line-of-sight checks with ray tracing or voxel-based visibility models to improve accuracy in detecting blind spots, especially in complex environments. Dynamic Adaptation Mechanisms: Incorporate reinforcement learning or deep learning techniques to allow the algorithm to learn from environmental interactions and adapt to changing conditions in real time. Redundancy and Fault Tolerance: Design partial redundancy into the deployment strategy to ensure robustness against node failures, maintaining high coverage even under partial network degradation. These enhancements will significantly improve the practicality and applicability of our approach, making it more aligned with real-world WSN deployment requirements..

## 5. Conclusion and future work

In this paper, we present a new Multi-objective Dung Beetle Optimization Algorithm (MODBO) inspired by the habits of dung beetles. Several features are introduced in MODBO that make it effective in solving multi-objective problems. In order to make the DBO algorithm capable of solving multi-objective problems, we first introduce non-dominated sorting, which is an algorithm that performs a quick sorting of multiple objectives to quickly categorize the solutions in order to obtain the values of the Pareto Front where each solution is located, with these values we can keep the solutions with the front values to participate in the search for the true Pareto front value in subsequent searches. Secondly, we introduce the competition mechanism and the neighborhood mechanism, which are introduced to solve the problem of the MODBO algorithm relying heavily on the global optimum and local optimum values during the search process. Competitive mechanisms can guide the particle position update while maintaining good diversity. The neighborhood mechanism is designed to enhance particles' local search ability, allowing each particle to refer to the information of particles within its neighborhood only. This article utilizes 24 test functions from CEC2020 to compare 9 algorithms, and all MODBO algorithms show excellent performance. This paper evaluates the performance of the MODBO algorithm in solving multi-objective optimization problems using quantitative and qualitative methods. Quantitatively, MODBO was assessed using the Inverted Generational Distance (IGD) metric, which measures convergence and distribution. The results showed that MODBO outperformed other algorithms in these aspects. Qualitatively, the Pareto optimal front was analyzed using Hypervolume (HV) metrics and visual graphs, demonstrating superior coverage and diversity of solutions. To further validate its practical effectiveness, we applied MODBO to a multi-objective sensor deployment problem, aiming to maximize coverage while minimizing the number of sensor nodes. The results confirmed that MODBO efficiently solved this engineering challenge, achieving high coverage with fewer sensors. This application highlights the algorithm's capability to handle complex multi-objective problems in real-world scenarios. In summary, the evaluation confirms that MODBO is highly effective in solving multi-objective optimization problems, excelling in both theoretical metrics and practical applications.

In future research, the algorithm can add some dynamic strategies to solve the dynamic processing ability of the algorithm after the environment changes. In sensor deployment, we will incorporate more constraints, such as energy

consumption limits, communication range restrictions, and physical space limitations. Additionally, the complexity of real-world application scenarios must be considered, including sensor node failure rates, the impact of environmental noise on data acquisition accuracy, and signal transmission stability under different terrains. By introducing adaptive adjustment mechanisms and distributed optimization techniques, sensor networks can achieve greater robustness and flexibility when facing dynamic changes. At the same time, by integrating machine learning methods, the algorithm's predictive capabilities for unknown environments and decision-making efficiency can be further improved, leading to more efficient resource utilization and task execution.

## Supporting information

**S1 File. Coded file.**
(ZIP)

## Author contributions

**Formal analysis:** Lianhai Lin.

**Software:** Wenxing Wu.

**Supervision:** Wenxing Wu.

**Visualization:** Junyi Wu.

**Writing – original draft:** Wenxing Wu.

**Writing – review & editing:** Liqin Tian.

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
