## [Decision Letter · Decision Letter 0]

16 Jun 2024

Dear Dr. Wu,

Thank you for submitting your manuscript to PLOS ONE. After careful consideration, we feel that it has merit but does not fully meet PLOS ONE’s publication criteria as it currently stands. Therefore, we invite you to submit a revised version of the manuscript that addresses the points raised during the review process.

We look forward to receiving your revised manuscript.

Kind regards,

Alireza Goli

Academic Editor

PLOS ONE

Journal Requirements:

Additional Editor Comments:

Reviewer 1:

Does the proposed Multi-Objective Dung Beetle Optimization (MODBO) algorithm incorporate competitive and neighborhood mechanisms for solving multi-objective problems?

How does the introduction of fast, non-dominated sorting enhance the Dung Beetle Optimization Algorithm's ability to solve multi-objective optimization problems?

What role does the Competition mechanism play in guiding global optimal search within the MODBO algorithm?

In the introduction, you need to connect the state of the art to your paper goals. Please follow the literature review by a clear and concise state of the art analysis. This should clearly show the knowledge gaps identified and link them to your paper goals. Please reason both the novelty and the relevance of your paper goals. Clearly discuss what the previous studies that you are referring to. What are the Research Gaps/Contributions? Please note that the paper may not be considered further without a clear research gap and novelty of the study.

Literature Review has the chance to be further improved: it seems that the authors have made the retrospection. However, via the review, what issues should be addressed? What is the current specific knowledge gap? What implication can be referred to? The above questions should be answered. Authors need to propose their study and compare it with A Multi Echelon Location-Routing-Inventory Model for a Supply Chain Network: NSGA II and Multi-Objective Whale Optimization Algorithm, A new modified social engineering optimizer algorithm for engineering applications

How does the Neighborhood mechanism assist in guiding local optimal value search in the MODBO algorithm?

What purpose does the external archive serve within the MODBO algorithm, and how does it contribute to achieving optimality?

How was the effectiveness of the MODBO algorithm evaluated, and what specific benchmarks were used in comparison with other algorithms?

Can you explain how the MODBO algorithm performed when tested on engineering problems, such as the 3D sensor deployment problem?

What are the key findings from the comparison of MODBO with other algorithms in terms of solving complex multi-objective problems, both in terms of convergence and distribution

Reviewer 2:

The article titled "Multi-objective dung beetle optimization algorithm: A novel algorithm for solving complex multi-objective optimization problems" introduces a new optimization algorithm, MODBO, which builds upon the existing dung beetle optimization algorithm (DBO) to address multi-objective problems (MOPs). This novel algorithm incorporates mechanisms such as competitive and neighborhood mechanisms, non-dominated sorting, and external archiving to enhance its capability to find optimal solutions in complex multi-objective spaces.

Key Components and Innovations:

1. Non-Dominated Sorting and External Archiving

Non-dominated sorting is a critical technique in multi-objective optimization that classifies solutions based on Pareto dominance. Solutions that are not dominated by any other solution form a Pareto front. MODBO integrates non-dominated sorting to efficiently manage and evolve the population of solutions. Additionally, an external archive is used to store the best non-dominated solutions found so far, ensuring that the algorithm maintains a diverse set of high-quality solutions across iterations.

2. Competition and Neighborhood Mechanisms

To enhance the search capabilities of the DBO algorithm, MODBO introduces two significant mechanisms:

• Competition Mechanism: This guides particles towards the global optimal solution by encouraging competition among particles. It helps in maintaining the diversity of the solutions and prevents premature convergence to suboptimal solutions.

• Neighborhood Mechanism: This mechanism focuses on local optimization by allowing particles to explore their immediate surroundings more thoroughly. It ensures that the algorithm can fine-tune solutions to achieve better local optimality.

3. Performance Evaluation and Benchmarking

The MODBO algorithm's performance is evaluated against nine established algorithms using the CEC2020 benchmark suite. Additionally, a practical application of MODBO is demonstrated through the 3D sensor deployment problem, which showcases its effectiveness in real-world scenarios.

Strengths:

1. Innovative Mechanisms for Improved Search

The incorporation of competition and neighborhood mechanisms addresses common issues in optimization algorithms, such as maintaining a balance between exploration and exploitation. These mechanisms ensure that the algorithm does not get trapped in local optima and can explore the solution space more effectively.

2. Comprehensive Benchmarking

The use of CEC2020 benchmark problems provides a robust and standardized means of evaluating the algorithm's performance. This comparison with established algorithms highlights the strengths and areas of improvement for MODBO.

3. Practical Application Demonstration

Applying MODBO to the 3D sensor deployment problem illustrates its practical utility and effectiveness in solving real-world problems. This practical validation adds credibility to the algorithm's theoretical advancements.

Weaknesses:

1. Complexity of Mechanisms

While the competition and neighborhood mechanisms enhance the algorithm's performance, they also add complexity. Implementing and tuning these mechanisms might require significant computational resources and expert knowledge, which could limit the algorithm's accessibility and usability.

2. Limited Scope of Applications Demonstrated

Although the 3D sensor deployment problem is a valid application, demonstrating MODBO's effectiveness across a wider range of practical problems would strengthen the claims about its versatility and robustness.

3. Dependence on Parameters

Like many optimization algorithms, MODBO's performance is highly dependent on the proper setting of various parameters (e.g., competition and neighborhood coefficients). Finding the optimal parameters can be challenging and may require extensive experimentation.

Suggestions for Improvement:

1. Simplification of Mechanisms

Exploring ways to simplify the competition and neighborhood mechanisms without significantly compromising performance could make the algorithm more accessible. For instance, adaptive mechanisms that automatically adjust parameters during the search process could reduce the need for manual tuning.

2. Broader Application Testing

Extending the evaluation of MODBO to a wider range of real-world problems would provide more evidence of its generalizability and robustness. Including diverse applications from different domains such as logistics, finance, and bioinformatics would strengthen the case for MODBO's versatility.

3. Parameter Sensitivity Analysis

Conducting a detailed sensitivity analysis of the algorithm's parameters would provide valuable insights into their impact on performance. This could help in developing guidelines or heuristics for setting parameters, making the algorithm more user-friendly.

4. Integration with Machine Learning Techniques

Integrating MODBO with machine learning techniques could further enhance its performance. For example, using machine learning to predict the effectiveness of certain parameter settings or to guide the search process could improve both the efficiency and the outcomes of the optimization.

Conclusion:

The Multi-Objective Dung Beetle Optimization Algorithm (MODBO) represents a significant advancement in the field of multi-objective optimization. By leveraging innovative mechanisms and robust benchmarking, it demonstrates strong potential for solving complex optimization problems. However, addressing its complexity, expanding its application scope, and optimizing its parameters could further enhance its usability and effectiveness.

Reviewers' comments:

Reviewer's Responses to Questions

**Comments to the Author**

1. Is the manuscript technically sound, and do the data support the conclusions?

Reviewer #1: Yes

Reviewer #2: Yes

2. Has the statistical analysis been performed appropriately and rigorously?

Reviewer #1: No

Reviewer #2: Yes

3. Have the authors made all data underlying the findings in their manuscript fully available?

Reviewer #1: Yes

Reviewer #2: Yes

4. Is the manuscript presented in an intelligible fashion and written in standard English?

Reviewer #1: Yes

Reviewer #2: Yes

Reviewer #1: Does the proposed Multi-Objective Dung Beetle Optimization (MODBO) algorithm incorporate competitive and neighborhood mechanisms for solving multi-objective problems?

How does the introduction of fast, non-dominated sorting enhance the Dung Beetle Optimization Algorithm's ability to solve multi-objective optimization problems?

What role does the Competition mechanism play in guiding global optimal search within the MODBO algorithm?

In the introduction, you need to connect the state of the art to your paper goals. Please follow the literature review by a clear and concise state of the art analysis. This should clearly show the knowledge gaps identified and link them to your paper goals. Please reason both the novelty and the relevance of your paper goals. Clearly discuss what the previous studies that you are referring to. What are the Research Gaps/Contributions? Please note that the paper may not be considered further without a clear research gap and novelty of the study.

Literature Review has the chance to be further improved: it seems that the authors have made the retrospection. However, via the review, what issues should be addressed? What is the current specific knowledge gap? What implication can be referred to? The above questions should be answered. Authors need to propose their study and compare it with A Multi Echelon Location-Routing-Inventory Model for a Supply Chain Network: NSGA II and Multi-Objective Whale Optimization Algorithm, A new modified social engineering optimizer algorithm for engineering applications

How does the Neighborhood mechanism assist in guiding local optimal value search in the MODBO algorithm?

What purpose does the external archive serve within the MODBO algorithm, and how does it contribute to achieving optimality?

How was the effectiveness of the MODBO algorithm evaluated, and what specific benchmarks were used in comparison with other algorithms?

Can you explain how the MODBO algorithm performed when tested on engineering problems, such as the 3D sensor deployment problem?

What are the key findings from the comparison of MODBO with other algorithms in terms of solving complex multi-objective problems, both in terms of convergence and distribution

Reviewer #2: The article titled "Multi-objective dung beetle optimization algorithm: A novel algorithm for solving complex multi-objective optimization problems" introduces a new optimization algorithm, MODBO, which builds upon the existing dung beetle optimization algorithm (DBO) to address multi-objective problems (MOPs). This novel algorithm incorporates mechanisms such as competitive and neighborhood mechanisms, non-dominated sorting, and external archiving to enhance its capability to find optimal solutions in complex multi-objective spaces.

Key Components and Innovations:

1. Non-Dominated Sorting and External Archiving

Non-dominated sorting is a critical technique in multi-objective optimization that classifies solutions based on Pareto dominance. Solutions that are not dominated by any other solution form a Pareto front. MODBO integrates non-dominated sorting to efficiently manage and evolve the population of solutions. Additionally, an external archive is used to store the best non-dominated solutions found so far, ensuring that the algorithm maintains a diverse set of high-quality solutions across iterations.

2. Competition and Neighborhood Mechanisms

To enhance the search capabilities of the DBO algorithm, MODBO introduces two significant mechanisms:

• Competition Mechanism: This guides particles towards the global optimal solution by encouraging competition among particles. It helps in maintaining the diversity of the solutions and prevents premature convergence to suboptimal solutions.

• Neighborhood Mechanism: This mechanism focuses on local optimization by allowing particles to explore their immediate surroundings more thoroughly. It ensures that the algorithm can fine-tune solutions to achieve better local optimality.

3. Performance Evaluation and Benchmarking

The MODBO algorithm's performance is evaluated against nine established algorithms using the CEC2020 benchmark suite. Additionally, a practical application of MODBO is demonstrated through the 3D sensor deployment problem, which showcases its effectiveness in real-world scenarios.

Strengths:

1. Innovative Mechanisms for Improved Search

The incorporation of competition and neighborhood mechanisms addresses common issues in optimization algorithms, such as maintaining a balance between exploration and exploitation. These mechanisms ensure that the algorithm does not get trapped in local optima and can explore the solution space more effectively.

2. Comprehensive Benchmarking

The use of CEC2020 benchmark problems provides a robust and standardized means of evaluating the algorithm's performance. This comparison with established algorithms highlights the strengths and areas of improvement for MODBO.

3. Practical Application Demonstration

Applying MODBO to the 3D sensor deployment problem illustrates its practical utility and effectiveness in solving real-world problems. This practical validation adds credibility to the algorithm's theoretical advancements.

Weaknesses:

1. Complexity of Mechanisms

While the competition and neighborhood mechanisms enhance the algorithm's performance, they also add complexity. Implementing and tuning these mechanisms might require significant computational resources and expert knowledge, which could limit the algorithm's accessibility and usability.

2. Limited Scope of Applications Demonstrated

Although the 3D sensor deployment problem is a valid application, demonstrating MODBO's effectiveness across a wider range of practical problems would strengthen the claims about its versatility and robustness.

3. Dependence on Parameters

Like many optimization algorithms, MODBO's performance is highly dependent on the proper setting of various parameters (e.g., competition and neighborhood coefficients). Finding the optimal parameters can be challenging and may require extensive experimentation.

Suggestions for Improvement:

1. Simplification of Mechanisms

Exploring ways to simplify the competition and neighborhood mechanisms without significantly compromising performance could make the algorithm more accessible. For instance, adaptive mechanisms that automatically adjust parameters during the search process could reduce the need for manual tuning.

2. Broader Application Testing

Extending the evaluation of MODBO to a wider range of real-world problems would provide more evidence of its generalizability and robustness. Including diverse applications from different domains such as logistics, finance, and bioinformatics would strengthen the case for MODBO's versatility.

3. Parameter Sensitivity Analysis

Conducting a detailed sensitivity analysis of the algorithm's parameters would provide valuable insights into their impact on performance. This could help in developing guidelines or heuristics for setting parameters, making the algorithm more user-friendly.

4. Integration with Machine Learning Techniques

Integrating MODBO with machine learning techniques could further enhance its performance. For example, using machine learning to predict the effectiveness of certain parameter settings or to guide the search process could improve both the efficiency and the outcomes of the optimization.

Conclusion:

The Multi-Objective Dung Beetle Optimization Algorithm (MODBO) represents a significant advancement in the field of multi-objective optimization. By leveraging innovative mechanisms and robust benchmarking, it demonstrates strong potential for solving complex optimization problems. However, addressing its complexity, expanding its application scope, and optimizing its parameters could further enhance its usability and effectiveness.

**Do you want your identity to be public for this peer review?** For information about this choice, including consent withdrawal, please see our Privacy Policy

Reviewer #1: No

Reviewer #2: No

---

## [Author Response · Author response to Decision Letter 1]

26 Jul 2024

[please see the details in "Response to Reviewers.doc"]

Thanks very much for reviewing our manuscript(Title: Multi-objective dung beetle optimization algorithm: A novel algorithm for solving complex multi-objective optimization problems). We appreciate all your comments and suggestions! In this revision, we have addressed a lot of these comments. We hope the revised manuscript has now met the publication standard of PLOS ONE.

We used the word to annotate the changes and uploaded an unannotated version.

In the file "Response to Reviewers.doc", Our point-to-point responses to the comments raised by the reviewers and the additional editor are listed

Thanks again.

---

## [Decision Letter · Decision Letter 1]

19 Aug 2024

Dear Dr. Tian,

Thank you for submitting your manuscript to PLOS ONE. After careful consideration, we feel that it has merit but does not fully meet PLOS ONE’s publication criteria as it currently stands. Therefore, we invite you to submit a revised version of the manuscript that addresses the points raised during the review process.

Reviewer 1:

Thank you for addressing all comments. The revisions improve the paper significantly. I appreciate your thorough response.

Reviewer 2:

Dear Author,

I have reviewed the paper you provided, and I believe a few areas could benefit from further refinement and clarification. While the article has seen some improvements, there are still opportunities to enhance the overall quality through thoughtful discussion and review of the technical and substantive aspects.

Specifically, I would suggest considering the incorporation of the following concepts, which appear to be relevant to the subject matter:

1. The combined use of whale and moth-flame optimization algorithms.

This approach could potentially offer refined optimization techniques that could strengthen the analytical framework.

2. An optimal task scheduling method in Fog-IoT networks, utilizing a combination of AO and Whale Optimization Algorithm (WOA).

This combination may provide valuable insights into improving the efficiency and effectiveness of task management within the Fog-IoT environment.

I believe the inclusion and thorough examination of these concepts would contribute to a more comprehensive and well-rounded article. By addressing these technical and distinguished issues, you can elevate the quality of the work and ensure that the audience gains a deeper understanding of the subject matter.

I am happy to discuss these suggestions in further detail or provide any additional support you may require. Please let me know if you have any questions or if there is anything else I can assist with.

Best regards,

We look forward to receiving your revised manuscript.

Kind regards,

Alireza Goli

Academic Editor

PLOS ONE

Journal Requirements:

Additional Editor Comments:

Reviewer 1:

Thank you for addressing all comments. The revisions improve the paper significantly. I appreciate your thorough response.

Reviewer 2:

Dear Author,

I have reviewed the paper you provided, and I believe a few areas could benefit from further refinement and clarification. While the article has seen some improvements, there are still opportunities to enhance the overall quality through thoughtful discussion and review of the technical and substantive aspects.

Specifically, I would suggest considering the incorporation of the following concepts, which appear to be relevant to the subject matter:

1. The combined use of whale and moth-flame optimization algorithms.

This approach could potentially offer refined optimization techniques that could strengthen the analytical framework.

2. An optimal task scheduling method in Fog-IoT networks, utilizing a combination of AO and Whale Optimization Algorithm (WOA).

This combination may provide valuable insights into improving the efficiency and effectiveness of task management within the Fog-IoT environment.

I believe the inclusion and thorough examination of these concepts would contribute to a more comprehensive and well-rounded article. By addressing these technical and distinguished issues, you can elevate the quality of the work and ensure that the audience gains a deeper understanding of the subject matter.

I am happy to discuss these suggestions in further detail or provide any additional support you may require. Please let me know if you have any questions or if there is anything else I can assist with.

Best regards,

Reviewers' comments:

Reviewer's Responses to Questions

**Comments to the Author**

Reviewer #1: All comments have been addressed

Reviewer #2: All comments have been addressed

2. Is the manuscript technically sound, and do the data support the conclusions?

Reviewer #1: Yes

Reviewer #2: Yes

3. Has the statistical analysis been performed appropriately and rigorously?

Reviewer #1: Yes

Reviewer #2: Yes

4. Have the authors made all data underlying the findings in their manuscript fully available?

Reviewer #1: No

Reviewer #2: Yes

5. Is the manuscript presented in an intelligible fashion and written in standard English?

Reviewer #1: Yes

Reviewer #2: Yes

Reviewer #1: Thank you for addressing all comments. The revisions improve the paper significantly. I appreciate your thorough response.

Reviewer #2: Dear Author,

I have reviewed the paper you provided, and I believe a few areas could benefit from further refinement and clarification. While the article has seen some improvements, there are still opportunities to enhance the overall quality through thoughtful discussion and review of the technical and substantive aspects.

Specifically, I would suggest considering the incorporation of the following concepts, which appear to be relevant to the subject matter:

1. The combined use of whale and moth-flame optimization algorithms.

This approach could potentially offer refined optimization techniques that could strengthen the analytical framework.

2. An optimal task scheduling method in Fog-IoT networks, utilizing a combination of AO and Whale Optimization Algorithm (WOA).

This combination may provide valuable insights into improving the efficiency and effectiveness of task management within the Fog-IoT environment.

I believe the inclusion and thorough examination of these concepts would contribute to a more comprehensive and well-rounded article. By addressing these technical and distinguished issues, you can elevate the quality of the work and ensure that the audience gains a deeper understanding of the subject matter.

I am happy to discuss these suggestions in further detail or provide any additional support you may require. Please let me know if you have any questions or if there is anything else I can assist with.

Best regards,

**Do you want your identity to be public for this peer review?** For information about this choice, including consent withdrawal, please see our Privacy Policy

Reviewer #1: No

Reviewer #2: No

---

## [Author Response · Author response to Decision Letter 2]

30 Aug 2024

Response #1: Thank you very much for your suggestion. We noticed a flaw in our article: most of the algorithms compared in the article are multi-objective versions of the original algorithms, ignoring that many of the algorithms have been richly improved, and their performance increases when they become multi-objective algorithms. So, we added an experiment to the article to reproduce the algorithm proposed in the literature [32], in which we combined the WOA and MFO algorithms in a multi-objective variant, after which it was tested on 24 test functions. The results show a significant improvement compared to the original algorithm, but there is a gap compared to our proposed algorithm.

We modified both Table 2 and Table 3 and modified the experimental plots Fig.6-Fig.11

Response #2: Thank you for your valuable comments and interest in our paper. You mentioned the idea of “AO and Whale Optimization Algorithm (WOA)-based Optimization Approach for Task Scheduling in Fog-IoT Networks.” We have less elaboration on this in Section IV, so we have added a description of this section at the beginning of Section IV, with corresponding references, and likewise, our contribution to sensor research. Our research focuses on the sensor deployment problem rather than task scheduling in Fog-IoT. Our work focuses on how to study sensor deployment in complex 3D environments using the MODBO algorithm for more efficient data collection and monitoring. We believe this approach can better utilize the performance of sensors and applies to real-world scenarios.

---

## [Decision Letter · Decision Letter 2]

3 Sep 2024

Dear Dr. Tian,

Thank you for submitting your manuscript to PLOS ONE. After careful consideration, we feel that it has merit but does not fully meet PLOS ONE’s publication criteria as it currently stands. Therefore, we invite you to submit a revised version of the manuscript that addresses the points raised during the review process.

Editor:

please consider the following comments:

1. Page 3: The introduction of the term "multi-objective" could be clarified earlier in the text to enhance reader understanding.

2. Page 4: The phrase "leveraging the effectiveness of the dung beetle optimization algorithm" should be rephrased for clarity and conciseness.

3. Page 5: Consider providing a brief explanation of the "Competition mechanism" and "Neighborhood mechanism" to ensure all readers grasp their significance.

4. Page 6: There is a typographical error in the sentence discussing "non-dominated sorting"; please correct it to maintain professionalism.

5. Page 7: The reference to "CEC2020" should include a brief description of what it entails for readers unfamiliar with it.

6. Page 8: The transition between sections could be smoother; consider adding a sentence to link the discussion of algorithms to the results presented.

7. Page 9: The table formatting in Table 2 is inconsistent; ensure all columns are aligned for better readability.

8. Page 10: The experimental results could benefit from a more detailed explanation of the metrics used for evaluation.

9. Page 11: The conclusion could be strengthened by summarizing the implications of the findings more explicitly.

10. Page 12: Ensure all figures are referenced in the text before they appear to guide the reader effectively through the manuscript.

We look forward to receiving your revised manuscript.

Kind regards,

Alireza Goli

Academic Editor

PLOS ONE

Journal Requirements:

Additional Editor Comments:

Editor:

please consider the following comments:

1. Page 3: The introduction of the term "multi-objective" could be clarified earlier in the text to enhance reader understanding.

2. Page 4: The phrase "leveraging the effectiveness of the dung beetle optimization algorithm" should be rephrased for clarity and conciseness.

3. Page 5: Consider providing a brief explanation of the "Competition mechanism" and "Neighborhood mechanism" to ensure all readers grasp their significance.

4. Page 6: There is a typographical error in the sentence discussing "non-dominated sorting"; please correct it to maintain professionalism.

5. Page 7: The reference to "CEC2020" should include a brief description of what it entails for readers unfamiliar with it.

6. Page 8: The transition between sections could be smoother; consider adding a sentence to link the discussion of algorithms to the results presented.

7. Page 9: The table formatting in Table 2 is inconsistent; ensure all columns are aligned for better readability.

8. Page 10: The experimental results could benefit from a more detailed explanation of the metrics used for evaluation.

9. Page 11: The conclusion could be strengthened by summarizing the implications of the findings more explicitly.

10. Page 12: Ensure all figures are referenced in the text before they appear to guide the reader effectively through the manuscript.

Reviewers' comments:

Reviewer's Responses to Questions

**Comments to the Author**

Reviewer #2: All comments have been addressed

2. Is the manuscript technically sound, and do the data support the conclusions?

Reviewer #2: Yes

3. Has the statistical analysis been performed appropriately and rigorously?

Reviewer #2: I Don't Know

4. Have the authors made all data underlying the findings in their manuscript fully available?

Reviewer #2: No

5. Is the manuscript presented in an intelligible fashion and written in standard English?

Reviewer #2: Yes

Reviewer #2: (No Response)

**Do you want your identity to be public for this peer review?** For information about this choice, including consent withdrawal, please see our Privacy Policy

Reviewer #2: No

---

## [Author Response · Author response to Decision Letter 3]

9 Oct 2024

Comment #1: Page 3: The introduction of the term "multi-objective" could be clarified earlier in the text to enhance reader understanding.

Response #1: Thanks to your comments, we have included an explanation of multi-targeting at the top of the abstract so that the reader understands the work of our article from the very beginning of reading it.

Comment #2: Page 4: The phrase "leveraging the effectiveness of the dung beetle optimization algorithm" should be rephrased for clarity and conciseness.

Response #2: Thank you for your comment. We have reworded the sentence, and the revised sentence reads, " Thanks to the dung beetle optimization algorithm's fast convergence and robust optimization finding ability in single-objective optimization algorithms."

Comment #3: Page 5: Consider providing a brief explanation of the "Competition mechanism" and "Neighborhood mechanism" to ensure all readers grasp their significance.

Response #3: Thanks to your comment, we have added an explanation of the two mechanisms at the beginning of section 2.

In this section, we explain the DBO algorithm and optimize the MODBO algorithm using the competition and neighborhood mechanisms. The purpose of using the competition mechanism is that the DBO algorithm relies heavily on the global and local optimums during the optimization process. Still, in a multi-objective optimization problem, the global optimums and local optimums are multiple vectors (Pareto front of the solutions of 1), and different solutions will have different effects on the particle guidance to ensure the diversity of the population; we choose the competition mechanism to guide the update of particle positions, which can be used to provide the diversity of the population. Different solutions will have different effects on particle guidance; to ensure the diversity of the population, we choose the competition mechanism to guide the update of particle positions, which can provide to the greatest extent that the particles will not converge prematurely and fall into the local optimum when they are approaching the best Pareto front. The neighborhood mechanism is proposed to improve the algorithm's robustness and ensure the uniformity of the solution because even if the search in some regions is not practical, the particles in the other areas can continue searching for possible solutions, increasing the possibility of finding the global optimal solution. We describe both mechanisms in detail in Sections 2.2.1 and 2.2.2.

Comment #4: Page 6: There is a typographical error in the sentence discussing "non-dominated sorting"; please correct it to maintain professionalism.

Response #4: Thanks to your comment, we have corrected “non-dominated sorting” throughout the article to ensure uniformity of expression.

Comment #5: Page 7: The reference to "CEC2020" should include a brief description of what it entails for readers unfamiliar with it.

Response #5: Thanks to your comments, we have redone the explanation of CEC2020 at the beginning of section 3.

In this section, we perform several experiments to investigate the performance of MODBO in solving large-scale MOPs. Functional tests have been conducted in this section using 24 test functions from CEC2020, a collection of test functions specifically designed for evaluating multi-objective optimization algorithms. These test functions simulate various properties in real-world problems, such as nonlinearity, modality, rotation, translation, and noise, to comprehensively evaluate the capabilities of the algorithms. The CEC2020 test function set contains a series of problems with different dimensions, complexities, and properties to test the algorithms' convergence, diversity preservation, and robustness. These include 14 bi-objective test functions and ten tri-objective test functions. The MODBO algorithm is compared with one of the nine algorithms in Table 1, using PS and PF plots to visualize the algorithm's performance.

Comment #6: Page 8: The transition between sections could be smoother; consider adding a sentence to link the discussion of algorithms to the results presented.

Response #5: Thank you for your comment. We have reworked the connections between sentences to make the article smoother.

Comment #7: Page 9: The table formatting in Table 2 is inconsistent; ensure all columns are aligned for better readability.

Response #7: Thanks for your comment; we have reformatted Table 2 to ensure a better reading experience.

Comment #8: Page 10: The experimental results could benefit from a more detailed explanation of the metrics used for evaluation.

Response #8: Thanks to your comments, we have explained the evaluated metrics in more detail to show the experimental results.

Comment #9: Page 11: The conclusion could be strengthened by summarizing the implications of the findings more explicitly.

Response #9: Thanks to your comments, We have added the meaning of indicators to the discussion of the paper's experimental results, which can strengthen the conclusions more clearly.

Comment #10: Page 12: Ensure all figures are referenced in the text before they appear to guide the reader effectively through the manuscript.

Response #10: Thanks to your comments, We have validated all figures in the full text to ensure that all figures are cited in the text.

We greatly appreciate your excellent comments.

---

## [Decision Letter · Decision Letter 3]

22 Oct 2024

Dear Dr. Tian,

Thank you for submitting your manuscript to PLOS ONE. After careful consideration, we feel that it has merit but does not fully meet PLOS ONE’s publication criteria as it currently stands. Therefore, we invite you to submit a revised version of the manuscript that addresses the points raised during the review process.

We look forward to receiving your revised manuscript.

Kind regards,

Khan Bahadar Khan, Ph.D

Academic Editor

PLOS ONE

Journal Requirements:

Reviewers' comments:

Reviewer's Responses to Questions

**Comments to the Author**

Reviewer #3: (No Response)

Reviewer #4: (No Response)

2. Is the manuscript technically sound, and do the data support the conclusions?

Reviewer #3: Partly

Reviewer #4: Partly

3. Has the statistical analysis been performed appropriately and rigorously?

Reviewer #3: Yes

Reviewer #4: No

4. Have the authors made all data underlying the findings in their manuscript fully available?

Reviewer #3: Yes

Reviewer #4: Yes

5. Is the manuscript presented in an intelligible fashion and written in standard English?

Reviewer #3: No

Reviewer #4: No

Reviewer #3: I appreciate the opportunity to review this manuscript, which presents the Multi-Objective Dung Beetle Optimization Algorithm (MODBO). The authors propose a novel approach to addressing complex multi-objective optimization problems (MOPs) by enhancing the original Dung Beetle Optimization (DBO) algorithm. Below are my comments and suggestions for improvement:

Novelty and Contribution:

The integration of competitive and neighborhood mechanisms into MODBO is commendable. The comparative analysis against established algorithms adds considerable value to the study.

Methodological Rigor:

The methodology is generally solid and well detailed.

Literature Positioning:

The introduction provides a comprehensive overview of related works but lacks clarity in organization. Currently, it reads as a list rather than a cohesive narrative. A more structured presentation that clearly articulates how this study fits within the existing literature would strengthen the manuscript.

I must express my concern regarding the significant similarities between this manuscript and the foundational work titled "Dung beetle optimizer: a new meta-heuristic algorithm for global optimization." and derived works titled "Multi-Strategy Improved Dung Beetle Optimization Algorithm and Its Applications". Specific images (e.g., Image 1 and Image 3) and certain phrasings, such as the discussion surrounding the "No Free Lunch Theorem", page 3, and the description of the DBO algorithm, sec 2.1, reflect similar content and structure.

Given these overlaps, it is crucial that the authors clearly distinguish their contributions from this foundational work to avoid issues of dual publication or academic integrity.

Clarity and Writing Quality:

Overall, the writing is clear, but results and conclusions sections could benefit from reorganization to enhance readability. Emphasizing key findings in the conclusion would provide a clearer takeaway for readers and highlight the study’s significance.

Additionally, ensuring that all acronyms (e.g., MOEAs) are defined upon first use would improve accessibility for all readers. Typos are still present around the text.

Figures and Tables:

While the figures and tables are informative, some appear raw. I recommend highlighting aggregated results and moving detailed data to an appendix to improve readability. It would also be beneficial to ensure that all figures have descriptive captions and that axes are appropriately labeled.

Limitations:

Addressing potential limitations of the research in the conclusion would provide a more balanced view and further strengthen the manuscript.

Overall Recommendation

In conclusion, I believe this manuscript makes a valuable and solid contribution to the field of optimization algorithms and has great potential for practical applications. I recommend acceptance with minor revisions, as I am confident that addressing the points mentioned will significantly enhance the quality and clarity of the work.

Thank you for considering my comments. I appreciate the effort the authors have put into this research and look forward to seeing how it develops.

Reviewer #4: 1. There exists already MODBO using non dominated sorting algorithm in the literature. Go through the following link https://doi.org/10.1016/j.heliyon.2024.e37286. How you method is different and what is need of proposing the same method.

2. Instead of using CEC 2020 test suit can the authors use the recent test suits for instance CEC 2022.

3. The comparative study is messing in the manuscript with respect to the technique. It is advised to test and compare with the above reference if both are different.

4. There is non-parametric test provided in the manuscript.

**Do you want your identity to be public for this peer review?** For information about this choice, including consent withdrawal, please see our Privacy Policy

Reviewer #3: No

Reviewer #4: No

---

## [Author Response · Author response to Decision Letter 4]

25 Nov 2024

SUGGESTIONS FROM EDITOR

Editor: Please review your reference list to ensure that it is complete and correct. If you have cited papers that have been retracted, please include the rationale for doing so in the manuscript text or remove these references and replace them with relevant current references. Any changes to the reference list should be mentioned in the rebuttal letter that accompanies your revised manuscript. If you need to cite a retracted article, indicate the article’s retracted status in the References list and also include a citation and full reference for the retraction notice.

Response #1: We have checked all the references and ensured that the format of the references meets the requirements of PLOS ONE, and we have replaced any problematic references.

REVIEWER #3

Comment #1: The introduction provides a comprehensive overview of related works but lacks clarity in organization. Currently, it reads as a list rather than a cohesive narrative. A more structured presentation that clearly articulates how this study fits within the existing literature would strengthen the manuscript.

Response #1: Thank you for your comment. We have added a guiding narrative in the introduction to introduce our work and clarify the differences between existing work and our paper.

Comment #2: I must express my concern regarding the significant similarities between this manuscript and the foundational work titled "Dung beetle optimizer: a new meta-heuristic algorithm for global optimization." and derived works titled "Multi-Strategy Improved Dung Beetle Optimization Algorithm and Its Applications". Specific images (e.g., Image 1 and Image 3) and certain phrasings, such as the discussion surrounding the "No Free Lunch Theorem", page 3, and the description of the DBO algorithm, sec 2.1, reflect similar content and structure.Given these overlaps, it is crucial that the authors clearly distinguish their contributions from this foundational work to avoid issues of dual publication or academic integrity.

Response #2: Thank you for your comments. We have carefully read the two references you provided and can point out significant differences between our article and these two articles. 1. Our article focuses on solving multi-objective problems, and the strategies proposed serve multi-objective functions, while these two articles focus more on single-objective algorithms. If we limit the goal of solving problems in our article to a single objective, our proposed algorithm can solve single-objective problems. However, the algorithms proposed in these two articles cannot solve multi-objective problems 2. The engineering problems solved in our article mainly focus on deploying sensors in 3D environments and explain the problems faced in deployment, which are closely related to reality. The problems tested in these two articles are more common standard test datasets such as Welded Beam Design Issues and Reducer Design Issues 3. As we are all making changes to the original DBO algorithm, there may be some similarities in the description of the original algorithm. However, our improvements on the original algorithm are different. The improvement strategy proposed in our article focuses more on the algorithm handling complex multi-objective problems, while these two articles focus on handling single-objective problems.

Comment #3: Overall, the writing is clear, but results and conclusions sections could benefit from reorganization to enhance readability. Emphasizing key findings in the conclusion would provide a clearer takeaway for readers and highlight the study’s significance. Additionally, ensuring that all acronyms (e.g., MOEAs) are defined upon first use would improve accessibility for all readers. Typos are still present around the text.

Response #3: Thank you for your comments. We have revised the wording of the conclusion section to make it more straightforward. We have checked all abbreviations in the article and provided explanations for them.

Comment #4: While the figures and tables are informative, some appear raw. I recommend highlighting aggregated results and moving detailed data to an appendix to improve readability. It would also be beneficial to ensure that all figures have descriptive captions and that axes are appropriately labeled.

Response #4: Thank you for your comments. We have rewritten the titles of some charts to ensure that they are descriptive and explained in the article.

Comment #5: Addressing potential limitations of the research in the conclusion would provide a more balanced view and further strengthen the manuscript.

Response #5: Thank you for your comments. We added Limitations in Section 4.5, which elaborated on the remaining issues with our algorithm in the article and provided further plans for future research.

Reviewer #4:

Comment #1: There exists already MODBO using non dominated sorting algorithm in the literature. Go through the following link https://doi.org/10.1016/j.heliyon.2024.e37286. How you method is different and what is need of proposing the same method.

Response #1: Thank you for your comment on our article. There are some similarities in the methods used in both articles. We introduced a non-dominated sorting mechanism to improve the beetle optimization algorithm to multi-objective. However, the article �https://doi.org/10.1016/j.heliyon.2024.e37286) is a sorting method used by the ancient algorithm NSGA-II. However, this article uses a newly proposed non-dominated sorting method in recent years, which reduces time complexity and has a better sorting ability than that article.

Regarding algorithm improvement, our paper idea and article aim to solve the problem of the beetle algorithm quickly getting stuck in local optima. Four improvement strategies were used in that article using the CF series test functions. Our article used two improvement strategies, using MMF series test functions. CF series test functions mainly focus on multi-objective optimization problems with constraints, while MMF test functions focus on multi-objective optimization problems without constraints but with multimodal characteristics. CF test functions are closer to practical engineering applications, while MMF test functions are more commonly used in academic research to examine the performance of algorithms in complex search spaces. Our goals are mainly set for engineering problems. The article examines the flight capability of drones under various constrained conditions. In contrast, our article aims to improve the population's distribution ability to ensure the integrity of sensor coverage.

Comment #2: Instead of using CEC 2020 test suit can the authors use the recent test suits for instance CEC 2022.

Response #2: Thank you for your comment on our article. CEC2020 includes 24 test functions, including tests in various multi-objective scenarios, including functions with complex multimodal frontiers, multiple local optima, and discontinuous Pareto frontiers, which can comprehensively test the ability of our algorithm. CEC2022 is mainly used to test the performance of single-objective algorithms. At the same time, CEC2023 focuses more on testing dynamic multi-objective algorithms, which is also one of the tasks we will be doing soon.

Comment #3: The comparative study is messing in the manuscript with respect to the technique. It is advised to test and compare with the above reference if both are different.

Response #3: Thank you for your comments. In comparing algorithms, we selected MMF1, MMF14, and MMF14_a as the PS set tests. MMF1 mainly tests the convergence and distribution ability of the algorithm when dealing with continuous and uniformly distributed Pareto sets. MMF14 mainly tests the exploration ability and robustness of the algorithm when dealing with multimodal Pareto sets, especially whether the algorithm can effectively find and maintain multiple local Pareto sets. Further test the exploration ability and robustness of MMF14_a algorithm when dealing with more complex, multimodal Pareto sets, as well as its performance in maintaining diversity and avoiding premature convergence. In the PF test, we selected MMF1, MMF12, and MMF14. MMF1 mainly tests the convergence and distribution ability of the algorithm when generating a continuous and uniformly distributed Pareto front, mainly whether the algorithm can evenly distribute solutions on the Pareto front; MMF12 mainly tests the distribution ability and adaptability of algorithms when dealing with non-uniform Pareto fronts, mainly whether the algorithm can evenly distribute solutions in regions of different densities. MMF14 mainly tests the exploration ability and robustness of the algorithm in dealing with multimodal Pareto frontiers, especially whether the algorithm can effectively find and maintain multiple local Pareto frontiers and effectively switch and explore between different modes. Different test sets test the performance of different algorithms. In the above article, the author also used different testing functions when conducting different testing items, such as CF1, CF2, CF4, and CF6 when testing parameter sensitivity and CF2, 3, 5, and 7 when generating IGD.

Comment #4: There is non-parametric test provided in the manuscript.

Response #4: Thank you for your comments. We have added Table 5 to the article. The Wilcoxon test for MODBO and other MO competitors was conducted and explained in detail.

---

## [Decision Letter · Decision Letter 4]

1 Dec 2024

Dear Dr. Tian,

Thank you for submitting your manuscript to PLOS ONE. After careful consideration, we feel that it has merit but does not fully meet PLOS ONE’s publication criteria as it currently stands. Therefore, we invite you to submit a revised version of the manuscript that addresses the points raised during the review process.

We look forward to receiving your revised manuscript.

Kind regards,

Khan Bahadar Khan, Ph.D

Academic Editor

PLOS ONE

Journal Requirements:

Reviewers' comments:

Reviewer's Responses to Questions

**Comments to the Author**

Reviewer #3: (No Response)

Reviewer #4: All comments have been addressed

2. Is the manuscript technically sound, and do the data support the conclusions?

Reviewer #3: (No Response)

Reviewer #4: Partly

3. Has the statistical analysis been performed appropriately and rigorously?

Reviewer #3: (No Response)

Reviewer #4: No

4. Have the authors made all data underlying the findings in their manuscript fully available?

Reviewer #3: (No Response)

Reviewer #4: No

5. Is the manuscript presented in an intelligible fashion and written in standard English?

Reviewer #3: (No Response)

Reviewer #4: Yes

Reviewer #3: Thank you for your responses to my previous comments. However, I find that several of the revisions do not adequately address the concerns raised. Below, I provide a detailed assessment of your responses and additional feedback for improvement:

Comment 1: Organization of the Introduction

The introduction remains largely a list of related works without a cohesive structure or clear articulation of how this study fits into the broader context. While you mentioned adding a guiding narrative, I expected a more substantial restructuring of the section.

Further suggestion: The introduction should be reorganized, requiring changes to its structure. Focus on clearly highlighting gaps in the existing literature and explaining how your study addresses them.

Comment 2: Potential Overlaps with Foundational Work

The response acknowledges similarities with foundational works but does not address the issue of textual overlap or reliance on figures from prior studies.

Further suggestion: It is critical to either rephrase sections with significant textual similarity in your own words or explicitly cite the source within the text. For any figures derived from previous works, please provide alternative visual representations or clearly attribute them to the original source.

Comment 3: Clarity and Readability of Results and Conclusions

I do not see a significant improvement in the clarity or readability of the results and conclusions sections.

Comment 4: Research Limitations

The limitations described focus on the problem's difficulty but fail to address trade-offs or potential weaknesses in the algorithm itself. A critical self-assessment is missing, which undermines the balance of the discussion.

Further suggestion: Please provide a thorough discussion of the algorithm’s limitations, such as potential trade-offs, scalability, or computational constraints. A balanced evaluation will enhance the transparency and scientific rigor of your work.

Conclusion

While I appreciate the effort to address the comments, the revisions made thus far do not sufficiently improve the manuscript. I encourage you to carefully consider the above suggestions to strengthen the introduction, clarify your contributions, and enhance the overall readability and quality of the manuscript.

Reviewer #4: All Queries asked by the reviewer have been answered satisfactory by the authors. It is advised to keep the code publicly after acceptance.

**Do you want your identity to be public for this peer review?** For information about this choice, including consent withdrawal, please see our Privacy Policy

Reviewer #3: No

Reviewer #4: No

---

## [Author Response · Author response to Decision Letter 5]

8 Jan 2025

SUGGESTIONS FROM EDITOR

Editor: Please review your reference list to ensure that it is complete and correct. If you have cited papers that have been retracted, please include the rationale for doing so in the manuscript text or remove these references and replace them with relevant current references. Any changes to the reference list should be mentioned in the rebuttal letter that accompanies your revised manuscript. If you need to cite a retracted article, indicate the article’s retracted status in the References list and also include a citation and full reference for the retraction notice.

Response #1: We have checked all the references and ensured that the format of the references meets the requirements of PLOS ONE, and we have replaced any problematic references.

REVIEWER #3

Comment #1: The introduction remains largely a list of related works without a cohesive structure or clear articulation of how this study fits into the broader context. While you mentioned adding a guiding narrative, I expected a more substantial restructuring of the section.

Further suggestion: The introduction should be reorganized, requiring changes to its structure. Focus on clearly highlighting gaps in the existing literature and explaining how your study addresses them.

Response #1: We have taken your valuable suggestions to heart and, as a result, have thoroughly revised the introductory section. To enhance clarity and coherence, we have restructured the content to provide a more logical flow of ideas. Additionally, we have expanded on the guiding principles to illuminate the significance and context of our research. These revisions strengthen the manuscript and better highlight the contributions of our work.

Comment #2: Potential Overlaps with Foundational Work

The response acknowledges similarities with foundational works but does not address the issue of textual overlap or reliance on figures from prior studies.

Further suggestion: It is critical to either rephrase sections with significant textual similarity in your own words or explicitly cite the source within the text. For any figures derived from previous works, please provide alternative visual representations or clearly attribute them to the original source.

Response #2: Thanks to your suggestions, we have meticulously reviewed the article and resolved all controversial graphic elements based on the feedback. We also improved the language throughout the manuscript based on a detailed test report from the checking website. These changes ensured the uniqueness of our article.

Comment #3: Clarity and Readability of Results and Conclusions

I do not see a significant improvement in the clarity or readability of the results and conclusions sections.

Response #3: We are grateful for your insightful comments, which guided us in thoroughly revising the conclusion section. This revision now provides a detailed summary of our research findings and emphasizes the significance and impact of our work. We have ensured that the conclusion effectively highlights our study's contributions and underscores its value to the field.

Comment #4: Research Limitations

The limitations described focus on the problem's difficulty but fail to address trade-offs or potential weaknesses in the algorithm itself. A critical self-assessment is missing, which undermines the balance of the discussion.

Further suggestion: Please provide a thorough discussion of the algorithm’s limitations, such as potential trade-offs, scalability, or computational constraints. A balanced evaluation will enhance the transparency and scientific rigor of your work.

Conclusion

Response #4: We appreciate your constructive comments, which have prompted us to revise the Limitations section comprehensively. In this updated section, we now provide a more detailed account of the current limitations within the Algorithms section. We have carefully considered and articulated the potential weaknesses in our algorithms, aiming to offer a transparent and honest assessment of our work.

---

## [Decision Letter · Decision Letter 5]

27 Jan 2025

Dear Dr. Tian,

Thank you for submitting your manuscript to PLOS ONE. After careful consideration, we feel that it has merit but does not fully meet PLOS ONE’s publication criteria as it currently stands. Therefore, we invite you to submit a revised version of the manuscript that addresses the points raised during the review process.

We look forward to receiving your revised manuscript.

Kind regards,

Khan Bahadar Khan, Ph.D

Academic Editor

PLOS ONE

Journal Requirements:

Reviewers' comments:

Reviewer's Responses to Questions

**Comments to the Author**

Reviewer #3: All comments have been addressed

Reviewer #5: All comments have been addressed

2. Is the manuscript technically sound, and do the data support the conclusions?

Reviewer #3: Yes

Reviewer #5: Yes

3. Has the statistical analysis been performed appropriately and rigorously?

Reviewer #3: Yes

Reviewer #5: Yes

4. Have the authors made all data underlying the findings in their manuscript fully available?

Reviewer #3: Yes

Reviewer #5: Yes

5. Is the manuscript presented in an intelligible fashion and written in standard English?

Reviewer #3: No

Reviewer #5: Yes

Reviewer #3: I appreciate the detailed effort the authors have made to address my comments. The manuscript has improved substantially in terms of structure.

While the manuscript is now suitable for publication, I suggest a careful review of the English language to improve readability. There are some awkward phrasings and grammatical issues that could be polished. For instance, in the introduction:

"Although many algorithms have been proposed and performed well in many real-world problems, it is worth mentioning."

I recommend rephrasing sentences like this to enhance clarity.

Overall, the authors have addressed all my concerns, and I have no further objections.

Reviewer #5: • Introduction:

• The introduction is now more organized and provides better context. However, it would benefit from slightly more focus on how your proposed algorithm directly addresses the gaps in the existing methods.

• Algorithm Explanation:

• The explanations of the Competition and Neighborhood mechanisms are clear and well-structured. No further changes are needed here.

• Results and Discussion:

• The results section is comprehensive and the statistical analyses are robust. However, a brief discussion on why MODBO performs slightly worse in some cases would improve balance and clarity.

• Figures and Tables:

• The figures and tables are clear and well-presented. Ensure that all figure captions briefly explain their relevance to the results.

• Limitations and Future Work:

• The updated limitations section is much improved. No additional changes are needed, but you might briefly mention specific future applications of MODBO to strengthen the conclusion.

• Language and Style:

• The language has improved, but a few minor typos remain (e.g., "rolling shithouse" in Section 2.1.4). Please fix these to maintain professionalism.

**Do you want your identity to be public for this peer review?** For information about this choice, including consent withdrawal, please see our Privacy Policy

Reviewer #3: No

Reviewer #5: **Yes: ** Anas Amaireh

---

## [Author Response · Author response to Decision Letter 6]

15 Feb 2025

REVIEWER #3

Comment #1: Introduction: The introduction is now more organized and provides better context. However, it would benefit from slightly more focus on how your proposed algorithm directly addresses the gaps in the existing methods.

Response #1: Thank you for your suggestions. We have revised the content of the introduction again, adding a review of some related work and providing a detailed explanation of the necessity of our research. We also elaborated on how our work addresses existing deficiencies.

Comment #2: The results section is comprehensive, and the statistical analyses are robust. However, a brief discussion on why MODBO performs slightly worse in some cases would improve balance and clarity.

Response #2: We sincerely thank the reviewer for the constructive comments. Based on these suggestions, we have added a detailed explanation in the results analysis section to further clarify why our algorithm performed slightly worse on certain test functions, thereby enhancing the completeness and rigor of this study.

Comment #3: The updated limitations section is much improved. No additional changes are needed, but you might briefly mention specific future applications of MODBO to strengthen the conclusion.

Response #3: We appreciate your suggestions. In response, we have revised the section on future work, providing a more detailed exposition of our planned research content.

Comment #4: The language has improved, but a few minor typos remain (e.g., "rolling shithouse" in Section 2.1.4). Please fix these to maintain professionalism.

Response #4: We sincerely appreciate your valuable suggestions. In response, we have thoroughly checked all the expressions in the article and corrected any instances of spelling errors or improper phrasing.

---

## [Decision Letter · Decision Letter 6]

4 Mar 2025

Dear Dr. Tian,

Thank you for submitting your manuscript to PLOS ONE. After careful consideration, we feel that it has merit but does not fully meet PLOS ONE’s publication criteria as it currently stands. Therefore, we invite you to submit a revised version of the manuscript that addresses the points raised during the review process.

We look forward to receiving your revised manuscript.

Kind regards,

Khan Bahadar Khan, Ph.D

Academic Editor

PLOS ONE

Journal Requirements:

Reviewers' comments:

Reviewer's Responses to Questions

**Comments to the Author**

Reviewer #3: All comments have been addressed

Reviewer #5: (No Response)

2. Is the manuscript technically sound, and do the data support the conclusions?

Reviewer #3: Yes

Reviewer #5: Yes

3. Has the statistical analysis been performed appropriately and rigorously?

Reviewer #3: Yes

Reviewer #5: Yes

4. Have the authors made all data underlying the findings in their manuscript fully available?

Reviewer #3: Yes

Reviewer #5: Yes

5. Is the manuscript presented in an intelligible fashion and written in standard English?

Reviewer #3: Yes

Reviewer #5: Yes

Reviewer #3: The author’s previous response appears to be directed toward Reviewer 5, based on its content. Please ensure that responses are correctly addressed to maintain clarity in the review process.

Reviewer #5: The manuscript presents a novel Multi-objective Dung Beetle Optimization Algorithm (MODBO), extending the traditional Dung Beetle Optimization (DBO) to handle multi-objective problems. The proposed algorithm incorporates competition and neighborhood mechanisms to enhance global and local search capabilities, respectively. The authors validate MODBO on CEC2020 benchmark functions and apply it to a 3D wireless sensor deployment problem, demonstrating its effectiveness compared to nine existing algorithms. The study provides a strong theoretical foundation, a well-structured methodology, and a comprehensive performance evaluation.

While the work is innovative and impactful, there are several areas that require improvement before publication.

Major comments:

• MODBO outperforms most algorithms but is surpassed by NSGA-II and MO_Ring_PSO_SCD in some test cases. The paper attributes MO_Ring_PSO_SCD’s success to ring topology and congestion distance but lacks a detailed explanation of MODBO’s weaknesses. A deeper analysis of these cases is needed to understand its limitations and potential improvements.

• MODBO introduces competition and neighborhood mechanisms, but the impact of key hyperparameters (competition intensity, archive size, neighborhood radius) is not analyzed. A sensitivity study should be included to clarify their effects on performance.

• The explanation of MODBO’s workflow (Figure 5) is detailed but lacks pseudocode. Adding structured pseudocode would improve clarity and reproducibility.

• The limitations section is well-written but could suggest solutions. Future work should explore hybridization with deep learning for parameter tuning, adaptation to dynamic environments for real-time optimization, and parallelization techniques for large-scale problems.

Minor comments:

• Some sentences are awkwardly phrased or redundant. For example, "rolling shithouse" in Section 2.1.4 appears to be a typographical error and should be corrected. A thorough proofreading is needed to improve clarity and professionalism.

• Some figures, such as Figure 5, would benefit from clearer annotations and more descriptive captions. Figure descriptions should be revised to better explain their purpose.

• Some mathematical symbols and equations are used without proper definitions. All symbols and variables should be introduced before their first occurrence to ensure consistency.

The manuscript presents a significant contribution to multi-objective optimization. However, a more detailed justification of performance differences, parameter sensitivity analysis, and minor textual refinements are needed before acceptance.

**Do you want your identity to be public for this peer review?** For information about this choice, including consent withdrawal, please see our Privacy Policy

Reviewer #3: No

Reviewer #5: No

---

## [Author Response · Author response to Decision Letter 7]

7 Apr 2025

SUGGESTIONS FROM EDITOR

REVIEWER #3

Comment #1: MODBO outperforms most algorithms but is surpassed by NSGA-II and MO_Ring_PSO_SCD in some test cases. The paper attributes MO_Ring_PSO_SCD’s success to ring topology and congestion distance but lacks a detailed explanation of MODBO’s weaknesses. A deeper analysis of these cases is needed to understand its limitations and potential improvements.

Response #1: Thank you for your comment. In the paper, we provide a detailed explanation of the advantages of the MO_Ring_PSO_SCD algorithm. This algorithm introduces a unique crowding distance calculation method, which effectively maintains diversity and internal uniformity when handling three-objective discontinuous multimodal functions. Although our proposed MODBO algorithm also enhances diversity and uniformity through a neighborhood mechanism, it is less flexible when dealing with these three highly deceptive functions, resulting in suboptimal performance. During the improvement process of MODBO, we considered whether to incorporate a similar mechanism. However, testing revealed that this approach excessively enhances solution distribution in the later stages of the algorithm. While such an approach does improve MODBO's performance on these three specific functions, it significantly underperforms compared to the improvement strategy proposed in this paper when applied to most test functions. Therefore, we decided not to adopt this mechanism. Our chosen improvement strategy achieves near-optimal solutions, which are acceptable for most problems. In the future, we will explore additional enhancements to the neighborhood mechanism to improve its performance on this type of test function.

Comment #2: MODBO introduces competition and neighborhood mechanisms, but the impact of key hyperparameters (competition intensity, archive size, neighborhood radius) is not analyzed. A sensitivity study should be included to clarify their effects on performance.

Response #2: We appreciate your insightful comment In the introduction of both the competition and neighborhood mechanisms, there are no hyperparameters that require adjustment. Within the competition mechanism, selections are made from the Pareto optimal front obtained after each iteration. For each particle, it only needs to choose a guiding particle from among those closest to itself on the Pareto optimal front. Detailed explanations can be found in Section 2.2.2. Concerning the selection radius for competitive particles, this is not an issue since the Pareto optimal front changes with each iteration. The focus should be on whether there are suitable guiding particles available on this front. Regarding the neighborhood mechanism, particles are only influenced by their preceding and succeeding particles, without consideration for others, thus eliminating the concern over a neighborhood radius. The sole instance where hyperparameter settings impact algorithm performance is in the population size selection within the original DBO algorithm, which we have thoroughly tested and discussed in Fig. 2.

Comment #3: The explanation of MODBO’s workflow (Figure 5) is detailed but lacks pseudocode. Adding structured pseudocode would improve clarity and reproducibility.

Response #3: We appreciate your valuable feedback. In the final section detailing the algorithm flow, we utilized a flowchart accompanied by a detailed explanation to clearly convey the process. Introducing pseudocode in addition to these elements would have led to redundancy, as all three components—flowchart, explanation, and pseudocode—would essentially cover the same information. Recognizing this potential overlap, during the previous revisions of the paper, we decided to omit the pseudocode to streamline the presentation and avoid unnecessary repetition. We hope this approach enhances the clarity and readability of our work. Of course, we have also provided the pseudocode for your reference.

Algorithm 1 General framework of MODBO

Input: Population size (N_pop), maximum number of iterations (〖Max〗_t).

Output: Optimal solution.

Initialize algorithm parameters: Population size (N_pop), maximum number of iterations (〖Max〗_t), population (P_0)( randomly generated)

Calculate the initial optimal value using (P_0)

While (t≤〖Max〗_t) do

Obtain non-dominated solutions using Non-Dominated Sorting (NDS).

Calculate Crowding Distance (CD).

Compute (P_j) Calculate Crowding Distance.

CombineP_0andP_j:P=P_0∩P_j.

Calculate the solutions of the combined P.

Perform non-dominated sorting on P based on NDR and CD.

Replace the original population with the generated P.

t=t+1

End while

Comment #4: The limitations section is well-written but could suggest solutions. Future work should explore hybridization with deep learning for parameter tuning, adaptation to dynamic environments for real-time optimization, and parallelization techniques for large-scale problems.

Response #4: We sincerely appreciate your insightful comment. We have incorporated your suggestions into the limitations section of our paper. Numerous scholars have already integrated intelligent optimization algorithms with deep learning and successfully applied these methods in various fields. In our future work, we plan to combine the MODBO algorithm with reinforcement learning to address optimization challenges in dynamic environments. Specifically, we intend to introduce an environmental monitoring mechanism and a dynamic change mechanism to effectively tackle dynamic multi-objective optimization problems.

Comment #5: Some sentences are awkwardly phrased or redundant. For example, "rolling shithouse" in Section 2.1.4 appears to be a typographical error and should be corrected. A thorough proofreading is needed to improve clarity and professionalism. Some figures, such as Figure 5, would benefit from clearer annotations and more descriptive captions. Figure descriptions should be revised to better explain their purpose. Some mathematical symbols and equations are used without proper definitions. All symbols and variables should be introduced before their first occurrence to ensure consistency.

Response #5: Thank you for your insightful comment. We have refined ambiguous and redundant expressions throughout the manuscript, ensuring clarity and precision in our writing. A thorough review of the entire paper has been conducted to eliminate any potential ambiguities and correct problematic wording. Additionally, we have revised all figure captions and table titles to enhance their clarity and consistency. All mathematical symbols used in the paper have been meticulously checked, with detailed explanations provided at their first appearance. Furthermore, we ensured that all variables in the formulas are consistent and clearly defined.

---

## [Decision Letter · Decision Letter 7]

1 Jun 2025

Dear Dr. Tian,

Thank you for submitting your manuscript to PLOS ONE. After careful consideration, we feel that it has merit but does not fully meet PLOS ONE’s publication criteria as it currently stands. Therefore, we invite you to submit a revised version of the manuscript that addresses the points raised during the review process.

We look forward to receiving your revised manuscript.

Kind regards,

Khan Bahadar Khan, Ph.D

Academic Editor

PLOS ONE

Journal Requirements:

Reviewers' comments:

Reviewer's Responses to Questions

**Comments to the Author**

Reviewer #3: All comments have been addressed

Reviewer #6: (No Response)

2. Is the manuscript technically sound, and do the data support the conclusions?

Reviewer #3: Yes

Reviewer #6: Partly

3. Has the statistical analysis been performed appropriately and rigorously?

Reviewer #3: Yes

Reviewer #6: No

4. Have the authors made all data underlying the findings in their manuscript fully available?

Reviewer #3: Yes

Reviewer #6: No

5. Is the manuscript presented in an intelligible fashion and written in standard English?

Reviewer #3: Yes

Reviewer #6: Yes

Reviewer #3: All my concerns have been tackled in past reviews and I don't have any additional comment for this statement.

Reviewer #6: After a thorough review of the latest (7th) revision of the manuscript and careful cross-checking with the suggestions from the editor and Reviewer #3, I confirm that the authors have satisfactorily addressed all critical feedback. However, I have some more suggestions to improve the quality of the manuscript.

The paper presents a potentially impactful algorithm, but the presentation suffers from poor structure, redundancy, and weak articulation of novelty. A clearer motivation, concise and focused narrative, improved grammar, and critical engagement with related work would significantly enhance its scholarly quality. In introduction section, several paragraphs repeat similar information about algorithm types and challenges. Start with a strong motivation and define the research gap clearly. Then review related work critically rather than just descriptively. Instead of listing algorithms, explain trends and challenges in existing MOP algorithms, especially in local vs. global search strategies.

Revise the abstract for brevity and grammar. Focus on the key contributions, methodology, and results in a structured format: problem → method → results → implications.

In section II, phrases like “different solutions will have different effects on particle guidance” and “to ensure the diversity of the population” are mentioned multiple times with little added value. This should be streamlined for conciseness and precision.

Add formal update rules or pseudocode describing how particle positions are changed based on winners. Explain with clear criteria how the "smaller angle" is computed and why it is a good measure.

Clarify the update rule with a step-by-step description or equation. Also, explain why a ring topology was chosen over other neighborhood structures (e.g., global best, von Neumann, etc.).

Replace the current block with a numbered algorithm pseudocode box. Clearly label which steps correspond to which sub-behaviors of DBO (e.g., Ball Rolling → Step 4).

Provide algorithmic insight into why MODBO performs better on certain functions (e.g., MMF1–MMF5) and worse on others (e.g., MMF10 or MMF13). Discuss the impact of problem features (like modality, deception, or Pareto set geometry) on MODBO’s behavior.

Apply appropriate non-parametric statistical tests across benchmark problems. Include tables or plots highlighting significance markers (e.g., symbols to denote better/similar/worse performance).

Boldface or color-code the best-performing results for each benchmark function. Add a summary row or figure showing the number of wins/losses/ties of MODBO against competitors. Replace some tables with visual summaries like bar plots or radar charts to enhance readability.

Briefly justify why the chosen MODBO parameters (e.g., α, β, S) are suitable. If manual tuning was done, explain the tuning approach (e.g., grid search, trial-and-error). Clarify whether parameters for other algorithms were adopted from literature or optimized independently.

The introduction to Section 4 begins abruptly, without clearly establishing why sensor deployment is an important test case or how it challenges conventional optimization methods. Begin Section 4 with a concise problem statement and justification: why is 3D sensor deployment a suitable test for MODBO?

Related work is mentioned with minimal detail and without analytical depth or comparative evaluation. For example, how the swarm intelligence strategies used by others differ structurally from MODBO remains unclear.

Clearly define all variables and equations with proper mathematical notation and explain their significance.

Binary perception modeling is assumed (i.e., 0 or 1 perception), which ignores signal strength attenuation or probabilistic sensing models common in WSN literature. Consider replacing binary coverage models with probabilistic sensing functions, as these better reflect physical reality.

Coverage is considered purely geometrically, without integrating communication range, power limits, or data routing, which are central to realistic WSN deployment. Extend the model to include communication constraints or at least acknowledge them as future considerations.

Clarify how terrain surfaces are modeled and how the algorithm handles line-of-sight obstruction. Consider incorporating ray-tracing or voxel-based visibility models for more accurate detection of blind spots. Provide a formal algorithmic procedure (e.g., pseudocode or flowchart) for determining blind zones.

Equations (17) and (18) are stated without context. Why is coverage maximization and node minimization sufficient? Are other objectives (energy, lifetime, redundancy) not relevant?

The optimization problem lacks constraints typically found in real WSN deployments, such as deployment cost limits, maximum allowable latency, or maximum power usage. Discuss how trade-offs are handled (e.g., Pareto front analysis) and consider adding constraints like maximum allowable energy per node.

How is the 3D terrain generated? What kinds of obstacles or elevation variations are present? Provide detailed terrain descriptions and explain the simulation setup, including the number of runs, random seeds, and environmental parameters.

Only a single experiment appears to be conducted, comparing random placement and MODBO. There is no statistical validation, confidence intervals, or multiple trials. Present coverage results with statistical metrics, e.g., average ± standard deviation over multiple trials.

The figures are referenced (Figs. 13–15), but there's no in-depth discussion of spatial distribution, blind spots, or redundant coverage. Add comparative visualizations (e.g., heatmaps, 3D terrain maps) showing the distribution and performance of sensors.

Thorough proofreading and possibly consultation with a native English speaker or language editor is advised. Carefully proofread and typeset all equations and variables.

**Do you want your identity to be public for this peer review?** For information about this choice, including consent withdrawal, please see our Privacy Policy

Reviewer #3: No

Reviewer #6: **Yes: ** Vijay Govindarajan

---

## [Author Response · Author response to Decision Letter 8]

10 Jul 2025

SUGGESTIONS FROM EDITOR

REVIEWER #6

Comment #1: In section II, phrases like “different solutions will have different effects on particle guidance” and “to ensure the diversity of the population” are mentioned multiple times with little added value. This should be streamlined for conciseness and precision.

Response #1: We sincerely appreciate the reviewer’s insightful comment regarding the need to improve the clarity and precision of the discussion on population diversity and particle guidance. In response to this suggestion, we have carefully revised Section II to eliminate redundant repetitions (e.g., repeated mentions of “different solutions will have different effects on particle guidance” and “to ensure the diversity of the population”) and have refined the explanation of how the competition mechanism contributes to maintaining population diversity. Specifically, we now clearly define what is meant by "population diversity" in the context of our study — referring to the spatial distribution and coverage of non-dominated solutions in the objective space. We also provide a more precise explanation of how the competition mechanism enhances diversity by enabling particles to adaptively select leaders and explore underrepresented regions of the Pareto front. This helps prevent premature convergence and improves global search capabilities. Additionally, we have clarified the role of the neighborhood mechanism in promoting algorithmic robustness and solution uniformity through localized interactions among particles.We believe these revisions significantly enhance the clarity, conciseness, and academic rigor of the section. Thank you again for this valuable suggestion.

Comment #2: Add formal update rules or pseudocode describing how particle positions are changed based on winners. Explain with clear criteria how the "smaller angle" is computed and why it is a good measure.

Response #2: Thank you very much for your insightful comments and suggestions regarding the use of the angle as a metric in our competitive mechanism. We appreciate the opportunity to provide further clarification on this aspect.The choice of using the smaller angle between particles as a criterion for competition is grounded in both theoretical rationale and practical effectiveness. Here’s a detailed explanation:

1. Alignment with Objective Space: In multi-objective optimization, solutions are often represented in an objective space where each dimension corresponds to one of the objectives. The direction from one solution to another can be interpreted as a vector indicating improvement or degradation across these objectives. By selecting the particle with the smaller angle relative to the current particle P, we effectively choose the solution that offers the most aligned direction of improvement. This ensures that the particle moves towards regions of the Pareto front that are more likely to offer better trade-offs.

2. Promotion of Diversity: Utilizing angles helps maintain diversity within the population by encouraging exploration in underrepresented areas of the objective space. Particles are guided not only towards better solutions but also towards less crowded regions, thereby preventing premature convergence and promoting a more uniform coverage of the Pareto front.

Comment #3:Clarify the update rule with a step-by-step description or equation. Also, explain why a ring topology was chosen over other neighborhood structures (e.g., global best, von Neumann, etc.).

Response #3: We sincerely thank the reviewer for the insightful comment. In response to your suggestion, we have added pseudocode in Section 2.2.3 to clarify the update rules of the neighborhood mechanism. In MODBO, we adopt a ring topology-based neighborhood structure instead of commonly used alternatives such as the global best or Von Neumann topologies. This design enhances local search ability by allowing each particle to interact only with its immediate neighbors, which promotes exploitation within local regions and reduces the risk of premature convergence. Meanwhile, it helps maintain population diversity by limiting information sharing and preserving a more uniform distribution of solutions across the objective space. Additionally, the simplicity of the ring topology facilitates efficient implementation and supports distributed or parallel computation, offering good scalability. We believe this strategy contributes significantly to balancing exploration and exploitation in MODBO.

Comment #4: Replace the current block with a numbered algorithm pseudocode box. Clearly label which steps correspond to which sub-behaviors of DBO (e.g., Ball Rolling → Step 4).

Response #4: We would like to thank the reviewers for their valuable comments and suggestions. We have added pseudocode at the beginning of Section 2.2, which summarizes both the improved strategies and the behavior of the original DBO algorithm.

Comment #5:Provide algorithmic insight into why MODBO performs better on certain functions (e.g., MMF1–MMF5) and worse on others (e.g., MMF10 or MMF13). Discuss the impact of problem features (like modality, deception, or Pareto set geometry) on MODBO’s behavior.

Response #5: We would like to thank the reviewer for the insightful comment. MODBO does not perform worse on MMF10 and MMF13 — it simply does not achieve results as good as those of certain other algorithms. The main reason is that MO_RING_PSO_SCD was specially adapted for these problems, and it employs a unique concept of congestion distance, which demonstrates excellent performance on three-objective discontinuous multi-peak functions. It should be noted that expecting an algorithm to significantly outperform all others across all problem instances is not only unrealistic but also violates the "No Free Lunch" (NFL) theorem, which states that no single optimization algorithm can be universally superior for all types of problems. Therefore, MODBO achieving suboptimal yet still competitive results on some test functions is sufficient to demonstrate its overall effectiveness. If one wishes for MODBO to achieve the best performance on every single problem, specific design modifications tailored to each problem would be necessary. For example, in the case of MMF15_l, which contains a bi-level structure and a special Pareto Set (PS) layout, achieving a similar distribution would require specialized population management strategies — such as increasing the population size or introducing additional perturbation mechanisms — to maintain strong global search capabilities throughout the entire optimization process. However, such changes may come at the cost of reduced local search performance on other types of problems. Striking a balance between early-stage exploration and late-stage exploitation, while maintaining superior performance across all types of problems, is an extremely challenging — if not impossible — task.

Comment #6: Apply appropriate non-parametric statistical tests across benchmark problems. Include tables or plots highlighting significance markers (e.g., symbols to denote better/similar/worse performance).Boldface or color-code the best-performing results for each benchmark function. Add a summary row or figure showing the number of wins/losses/ties of MODBO against competitors. Replace some tables with visual summaries like bar plots or radar charts to enhance readability.

Response #6: We would like to thank the reviewer for the valuable comments and suggestions. In response, we have added a summary row in the table to show on how many test functions our algorithm outperforms the compared algorithms. Moreover, the original table already includes bold and black markings to highlight the test functions where MODBO performs better than other algorithms. We believe that both the table and the coverage plots together provide sufficient evidence of our algorithm’s performance. Specifically, the table offers quantitative results demonstrating the competitiveness of our method, while the coverage plots visually illustrate how closely the obtained PS and PF align with the true PF, further confirming the effectiveness and robustness of our approach.

Comment #7: Briefly justify why the chosen MODBO parameters (e.g., α, β, S) are suitable. If manual tuning was done, explain the tuning approach (e.g., grid search, trial-and-error). Clarify whether parameters for other algorithms were adopted from literature or optimized independently.

Response #7: We would like to thank the reviewer for the insightful comments and suggestions. Regarding the parameter settings of the MODBO algorithm, we first reimplemented the original DBO algorithm, which is a single-objective optimization method, based on its source paper. After adapting it to the multi-objective scenario, we conducted extensive experiments using grid search to determine the most suitable parameter combination for MODBO. Similarly, for all the other compared algorithms, we reimplemented them based on their respective reference papers and performed thorough parameter tuning before incorporating them into our experimental comparisons. This ensures that all algorithms are fairly evaluated with well-adjusted parameter settings.

Comment #8: The introduction to Section 4 begins abruptly, without clearly establishing why sensor deployment is an important test case or how it challenges conventional optimization methods. Begin Section 4 with a concise problem statement and justification: why is 3D sensor deployment a suitable test for MODBO?

Response #8: We would like to thank the reviewer for the constructive comments, which helped improve the clarity and motivation of this section. To demonstrate the practical effectiveness of MODBO in real-world scenarios, we apply it to the 3D sensor deployment problem — a complex and challenging optimization task with significant implications in wireless sensor networks (WSNs). This problem involves determining the optimal spatial distribution of sensors in a three-dimensional environment to simultaneously maximize coverage and minimize deployment cost (i.e., the number of sensor nodes). The 3D sensor deployment problem is particularly suitable for evaluating MODBO due to several reasons. First, it is inherently multi-objective, requiring a trade-off between two conflicting objectives: maximizing sensing coverage and minimizing deployment cost (i.e., the number of sensor nodes). Second, the search space is highly complex due to the 3D spatial arrangement and overlapping sensing ranges, which pose challenges for traditional optimization algorithms in maintaining both convergence and diversity. Third, this problem often exhibits multiple local optima, making global search capability and exploration–exploitation balance crucial — features that MODBO is specifically designed to address. By applying MODBO to this realistic and challenging scenario, we aim to validate not only its theoretical performance on benchmark functions but also its potential for solving practical engineering problems with real-world constraints.

Comment #9: Binary perception modeling is assumed (i.e., 0 or 1 perception), which ignores signal strength attenuation or probabilistic sensing models common in WSN literature. Consider replacing binary coverage models with probabilistic sensing functions, as these better reflect physical reality.

Response #9: We would like to sincerely thank the reviewer for the insightful and constructive comment regarding the choice of the coverage model in our study. In this work, we adopted the binary coverage model, where a target is considered either fully covered or not covered at all by a sensor. We acknowledge that this approach simplifies the physical reality by neglecting factors such as signal attenuation and sensing uncertainty, which are commonly modeled using probabilistic sensing functions in wireless sensor network (WSN) literature. However, the use of the binary model was a deliberate choice based on several considerations. First, it allows us to focus on the core optimization capabilities of MODBO—particularly its convergence and diversity performance—without introducing additional complexity from environmental noise or probabilistic uncertainty. Second, the binary model is widely used in foundational studies on multi-objective sensor deployment, making it easier to compare our results with existing approaches. Lastly, it offers significantly lower computational overhead, which is important for conducting large-scale and repeatable experiments. We fully agree with the reviewer that future work should incorporate more realistic probabilistic sensing models to better reflect practical scenarios. We plan to explore such extensions in our future research to further enhance the applicability and robustness of our algorithm in real-world environments.

Comment #10: Coverage is considered purely geometrically, without integrating communication range, power limits, or data routing, which are central to realistic WSN deployment. Extend the model to include communication constraints or at least acknowledge them as future considerations.

Response #10: We sincerely appreciate the reviewer’s valuable comment regarding the current limitations of our coverage model. Indeed, in this study, the coverage was modeled purely from a geometric perspective, focusing on spatial deployment and sensing range, without incorporating other critical aspects of real-world WSN deployments such as communication constraints, power limitations, or data routing mechanisms. The primary reason for adopting this simplified model was to isolate and evaluate the fundamental multi-objective optimization capabilities of MODBO — particularly its performance in balancing coverage maximization and node number minimization. By removing additional complexities, we aimed to provide a clear and reproducible benchmark for assessing the algorithm's behavior in a controlled environment. However, we fully agree with the reviewer that for practical applications, a more comprehensive model should include communication range and energy consumption considerations. In fact, extending the current framework to incorporate these factors is part of our ongoing and future research directions. We believe that integrating such constraints will further enhance the realism and applicability of our approach in actual WSN deployments. We have added a discussion of these limitations and potential extensions in the revised manuscript, to clarify the scope and future development of our work.

Comment #11: Clarify how terrain surfaces are modeled and how the algorithm handles line-of-sight obstruction. Consider incorporating ray-tracing or voxel-based visibility models for more accurate detection of blind spots. Provide a formal algorithmic procedure (e.g., pseudocode or flowchart) for determining blind zones.

Response #11: Thank you for your suggestion. We have added pseudocode (Algorithm 4) to provide a more detailed explanation of this part. Although ray tracing and voxel-based visibility models offer high accuracy and robustness in handling complex scenes, we chose a more simplified approach in this study for the following reasons: computational efficiency — ray tracing and voxel-based models typically require substantial computing resources and time, especially in large-scale deployment scenarios. Considering the real-time requirements and computational costs in practical applications, we opted for a lighter mathematical model; problem specificity — this study focuses on specific types of terrain and sensor deployment problems, where blind spots can be effectively identified through simple geometric relationships and equation solving. The performance of this method in such scenarios is already sufficiently strong, making it unnecessary to introduce more complex models; ease of implementation and debugging — compared to ray tracing and voxel-based models, the equation-solving approach is easier to implement and debug in MATLAB, facilitating rapid iteration and validation of algorithm effectiveness during development. In future research, we plan to further explore ray tracing and voxel-based visibility models to enhance the adaptability and accuracy of the algorithm in more complex environments.

Comment #

---

## [Decision Letter · Decision Letter 8]

20 Aug 2025

Multi-objective dung beetle optimization algorithm: A novel algorithm for solving complex multi-objective optimization problems

PONE-D-24-09818R8

Dear Dr. Tian,

We’re pleased to inform you that your manuscript has been judged scientifically suitable for publication and will be formally accepted for publication once it meets all outstanding technical requirements.

Kind regards,

Khan Bahadar Khan, Ph.D

Academic Editor

PLOS ONE

Additional Editor Comments (optional):

Reviewers' comments:

Reviewer's Responses to Questions

**Comments to the Author**

Reviewer #7: (No Response)

2. Is the manuscript technically sound, and do the data support the conclusions?

Reviewer #7: Yes

3. Has the statistical analysis been performed appropriately and rigorously?

Reviewer #7: Yes

4. Have the authors made all data underlying the findings in their manuscript fully available?

Reviewer #7: Yes

5. Is the manuscript presented in an intelligible fashion and written in standard English?

Reviewer #7: Yes

Reviewer #7: The manuscript is well-structured and presents novel contributions; therefore, it can now be accepted.

**Do you want your identity to be public for this peer review?** For information about this choice, including consent withdrawal, please see our Privacy Policy

Reviewer #7: No

---

## [Editor Report · Acceptance letter]

PONE-D-24-09818R8

PLOS ONE

Dear Dr. Tian,

I'm pleased to inform you that your manuscript has been deemed suitable for publication in PLOS ONE. Congratulations! Your manuscript is now being handed over to our production team.

Kind regards,

on behalf of

Dr. Khan Bahadar Khan

Academic Editor

PLOS ONE